# HiddenEcho: Mitigating Noise Amplification in Differentially Private LLMs with Hidden-State Correction

**Wenhao Li & Kunhao Li** *
School of Software Engineering
South China University of Technology
Guangzhou, China
`wenhaoli-lwh@outlook.com, kunhomlihf@gmail.com`

**Lei Yang** †
School of Software Engineering
South China University of Technology
Guangzhou, China
`sely@scut.edu.cn`

## Abstract

The rise of large language models (LLMs) has driven the adoption of Model-as-a-Service (MaaS). However, transmitting raw text to servers raises critical privacy concerns. Existing approaches employ deep neural networks (DNNs) or differential privacy (DP) to perturb inputs. Yet, these approaches suffer notable limitations: DNN-based methods often require task-specific pre-training, and conventional DP techniques, though privacy-preserving, suffer from noise amplification as perturbed inputs propagate through the deep transformer layer, leading to significant degradation in downstream task performance. To alleviate this, we propose `HiddenEcho`, an end-to-end framework with client noise correction, where hidden states are sent from the server to the client and refined by a lightweight module using both embeddings and intermediate representations. `HiddenEcho` suppresses inter-layer noise amplification without pretraining, effectively preserving task-relevant signals under DP constraints. To further reduce communication, `HiddenEcho` incorporates gradient-based hidden layer selection and information bottleneck compression, reducing communication cost while preserving essential task information. Experiments across text classification and generation tasks demonstrate that `HiddenEcho` achieves up to 46.89% performance improvement over DP baselines, over 85% communication reduction, and up to 72.52% faster training compared to existing denoising approaches, establishing a new privacy-utility trade-off for privatized LLMs. Codes are available at https://github.com/liwh011/hidden-echo.

## 1 Introduction

The advancement of large language models (LLMs) has profoundly transformed scientific research Kulmanov et al. (2024); VM et al. (2024); Li et al. (2023b); Yang et al. (2024b). The substantial computational costs associated with the growing number of parameters in LLMs have driven the emergence of the Model-as-a-Service (MaaS) paradigm. MaaS offers a platform for users without access to high-performance computing resources, enabling them to leverage LLMs for various purposes, including inference, fine-tuning, and the development of customized agents (David et al., 2014). Nevertheless, MaaS also raises significant security concerns. Specifically, sensitive in-

---

*Equal contributions.
†Corresponding author.

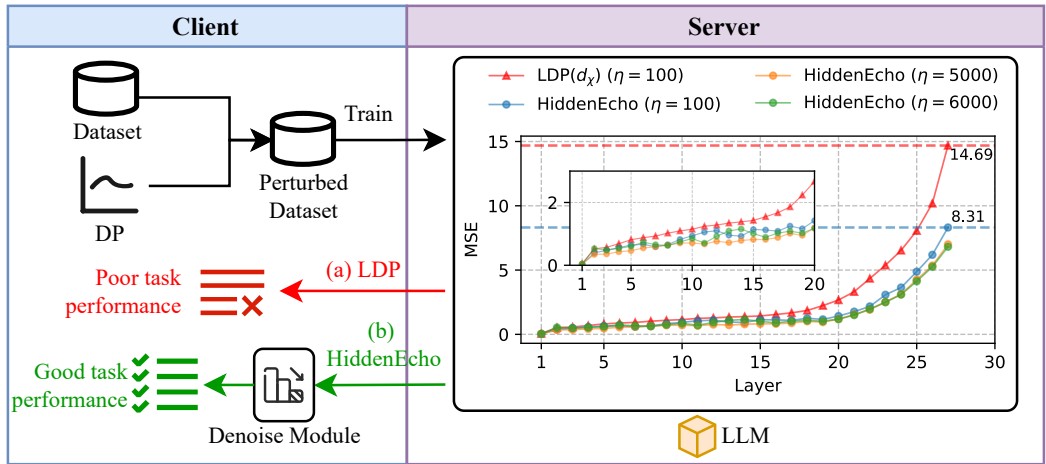

Figure 1: Mean squared error (MSE) between clean hidden states and noisy hidden states under different privacy budgets based on Qwen2-1.5B (Yang et al., 2024a) with 27 hidden layers retaining on the server side on the MRPC dataset (Wang et al., 2018).

formation, such as personally identifiable information (PII), including names, phone numbers, email addresses, and financial details, may be exposed when users upload data to LLM vendors.

Privacy protection for LLMs in the MaaS framework mainly relies on cryptography-based and perturbation-based methods. While cryptographic techniques like secure multiparty computation (Hou et al., 2024) and homomorphic encryption (Liu & Liu, 2023) provide strong security, their high computational overhead makes them impractical for resource-constrained clients.

In contrast, perturbation-based methods have gained attention because of their flexibility to add perturbations to the data as a privacy-preserving mechanism. For instance, deep neural network (DNN)-based perturbation methods leverage learned data distributions to generate perturbed data that can deceive adversaries. However, these approaches typically require pretraining phases for the whole training, limiting their practicality. Differential privacy (DP) as a perturbation-based method with lower computational overhead, has emerged as an alternative. It introduces noise of a specified intensity to the input on the client side before transmitting it to the server. For example, Qu *et al.* (Qu et al., 2021) proposed adding $d_\chi$-DP noise (based on $\chi^2$ distance to prevent reconstruction of the original data) to text embeddings, achieving enhanced privacy protection at the cost of reduced accuracy. However, when such noise is left unprocessed, it leads to significant performance degradation in downstream tasks when applied to LLM. Mai *et al.* (Mai et al., 2024) improve this issue with their SnD framework, which involves pretraining a denoising module on the server and deploying it on the client. This approach filters out a part of noises and enhances model performance.

Nevertheless, Experiments show that differential privacy noise in text embeddings is progressively amplified through LLM transformer blocks, leading to increasing MSE and significant performance degradation, as seen in the "LDP($d_\chi$) ($\eta$=100)" curve in Fig 1. Existing denoising methods, relying on pretraining and disconnected from LLM dynamics, fail to mitigate inter-layer noise effectively.

Based on this, we propose an end-to-end framework **HiddenEcho** that integrates noise correction in the MaaS to protect data privacy in LLMs. Unlike existing denoising approaches: (1) it eliminates the need for pretraining, enabling effective denoising of inter-layer noise from the server; (2) it fully leverages the internal hidden layer information of LLMs, optimizing their performance; and (3) Considering the communication overhead between the client and server, we introduce a gradient-based hidden layer filter to identify and select critical hidden layers, alongside an information bottleneck-based dimension reducer to retain essential information from the hidden states. This design enables near-complete noise correction with minimal data transmission, striking an effective balance between communication efficiency and model performance. As illustrated by the "HiddenEcho" curves in Fig 1, in the final hidden layer, HiddenEcho ($\eta$=100) reduces noise (14.69 → 8.31) by 43.43% compared to LDP($d_\chi$) ($\eta$=100).

In summary, our contributions are: ❶ We identify and analyze the critical issue of noise amplification in LLMs under differential privacy, where injected noise grows progressively through hidden layers, severely degrading model performance. ❷ We propose `HiddenEcho`, an end-to-end framework that enables pretraining-free, progressive noise correction via client-side denoising guided by hidden states from server, which is applicable to both inference and fine-tuning with balanced privacy, utility, and communication cost. ❸ We evaluate `HiddenEcho` in MaaS scenarios, showing up to 46.89% performance gain in text classification over baselines, over 85% communication reduction with `HiddenEcho`, and 72.52% faster denoising compared to existing methods.

## 2 RELATED WORKS

**Privacy Preservation for LLMs** Privacy preservation in LLMs has become critical with widespread deployment (Miranda et al., 2024). Existing approaches fall into cryptographic and perturbation-based methods. Cryptographic techniques, such as secure multi-party computation (Hou et al., 2024) and homomorphic encryption (Hao et al., 2022; Liu & Liu, 2023), offer strong privacy guarantees but incur high computational costs and are limited to defending against external adversaries, making them impractical for resource-constrained clients. Perturbation-based methods provide a more flexible trade-off between privacy and utility. While some approaches perturb model outputs (Liu et al., 2019) or use adversarial training (Coavoux et al., 2018a), differential privacy has emerged as a popular choice in the MaaS paradigm due to its lightweight noise injection into embeddings (Lyu et al., 2020; Qu et al., 2021; Shen et al., 2023; Li et al., 2023a). However, DP noise is amplified through transformer layers, degrading model performance. SnD (Mai et al., 2024) introduces a client-side denoising module to mitigate this effect, but fails to fully address noise propagation across deep transformer blocks—a challenge our work aims to resolve.

## 3 PRELIMINARIES

### 3.1 THREAT MODELS

For language models, attackers typically aim to extract sensitive information from the original user data. We consider a split MaaS deployment in which the client hosts the embedding layer and the server hosts the remaining model Shen et al. (2023). They follow the protocol but may attempt to infer additional information from observed artifacts. An attacker may be either (i) a malicious service provider), or (ii) an eavesdropper possessing any subset of the following: ❶ **Perturbed embeddings:** the attacker observes perturbed token embeddings $\Psi(x) = \mathcal{E}(x)) + \delta$ submitted by the client. ❷ **Embedding layer parameters:** the attacker observes the embedding matrix $W_{emb}$ used to map tokens to vectors. As highlighted in (Song & Raghunathan, 2020; Shen et al., 2023), Embedding Inversion Attacks (EIA) and Attribute Inference Attacks (AIA) represent significant privacy threats in machine learning:

**Definition 1 (Embedding Inversion Attack (EIA))** *Given perturbed embeddings* $\Psi(x) \in \mathbb{R}^{l \times d}$ *and the embedding matrix* $W_{emb}$, *the goal is to reconstruct each token t is recovered by*

$$\hat{v}_t = \arg\min_{v \in \mathcal{V}} \|W_{emb}[v] - \Psi(x)\|_2.$$

**Definition 2 (Attribute Inference Attack (AIA))** *Let* $a \in \mathcal{A}$ *be a sensitive attribute. Given auxiliary labeled samples* $\mathcal{S} = \{(\tilde{x}_i, \tilde{a}_i)\}$, *the attacker trains*

$$f_w : \mathbb{R}^{l \times d} \to \mathcal{A}$$

*on* $(\Psi(\tilde{x}_i), \tilde{a}_i)$ *and predicts* $\hat{a} = f_w(\Psi(x))$ *for target x.*

### 3.2 PROBLEM DEFINITION

Based on threat models, we focus on the privacy concerns associated with data transfer between the client and server when utilizing LLMs in the MaaS. In this scenario, the client holds a private dataset $X = \{x_1, x_2, \cdots, x_n\}$. Following a split learning framework Gupta & Raskar (2018); Zhang et al. (2023b), we mitigate the client's resource constraints by deploying the word embedding

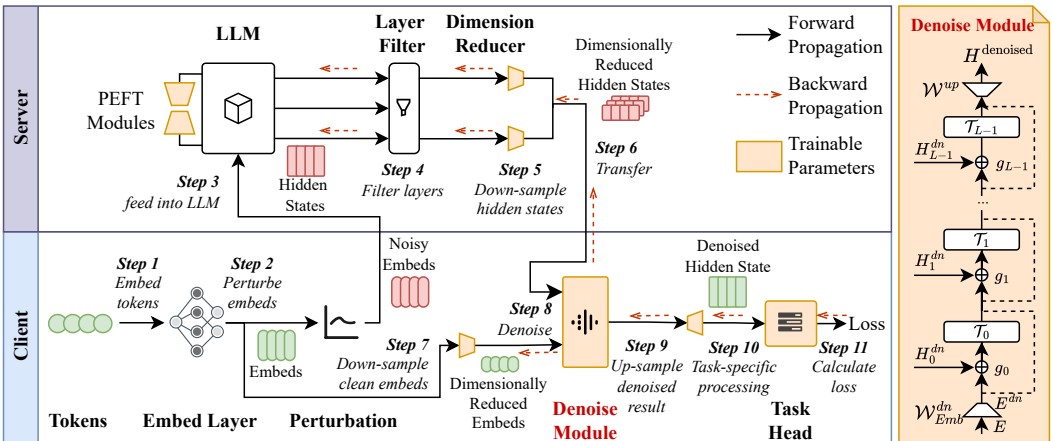

Figure 2: Framework of `HiddenEcho`. The denoise module is deployed on the client side, and the operations related to LLM's hidden layer of the Down-sampling, Layer Filter, and Dimension Reducer are deployed on the server side.

layer $\mathcal{E}$ of the LLM on the client, while the remaining layers are hosted on the server. To ensure privacy, perturbations based on differential privacy, denoted as $\delta$, are applied to the embeddings on the client. The optimization of the global LLM after incorporating these perturbations can be formalized as follows:

$$\theta^* = \arg\min_{\theta} \frac{1}{|X|} \sum_{x_i \in X} \mathcal{L}(\theta, \Psi(\mathcal{E}(x_i) + \delta)). \quad (1)$$

Here, $\theta$ are the model parameters to be optimized, and $\Psi$ denotes a denoising module. To enhance the feedback received by the client from the server, the design of an effective $\Psi$ for mitigating the impact of added noise on the model's outputs is crucial.

## 4 METHODOLOGY

`HiddenEcho` leverages a hidden layer correction to address noise amplification in LLMs. Under the split learning framework, `HiddenEcho` reduces transmission significantly with only a minor performance trade-off. Fig. 2 and Algorithm 1 provide detailed descriptions. The complexity analysis, theoretical justification, and comparison with DP are presented in Appendix D, F, G.

### 4.1 FULL NOISE CORRECTION

In `HiddenEcho`, server-side hidden layer states are transmitted back to the client for correction. This process is designed to be integrated with the fine-tuning of the LLM.

**Perturbation**  Tokenized texts are converted to embeddings $E = \mathcal{E}(x_i) \in \mathbb{R}^{n \times d}$ on the client, where $n$ is the sequence length and $d$ is the hidden size of the server-side LLM. To ensure privacy, noise is added to embeddings, yeilding $E' = E + \delta$, which are then transmitted to the server.

**Server-side Forward Propagation**  The server inputs the noisy embeddings $E'$ into the LLM $\mathcal{B}$. During forward propagation, intermediate hidden states $\boldsymbol{H} = \mathcal{B}(E') = \{H_0, \cdots, H_{L-1}\}$ are collected from all $L$ layers. However, injected noise progressively distorts the hidden states' feature space, which prevents LLM from effectively learning the task information. Consequently, a denoising mechanism is crucial to correct these hidden states for effective task learning.

**Denoising**  The client-side denoising module refines the hidden states received from the server. Drawing inspiration from the LST method (Sung et al., 2022), which uses a dimension-reduced LLM as a side network for downstream task learning, the denoising module takes the initial noise-free

embedding and the hidden states of the LLM on the server side as input. By utilizing the information contained in the initial embedding, it generates optimized hidden states: $H^{\text{denoised}} = \mathcal{D}(E, \boldsymbol{H})$, where $\mathcal{D}$ is the denoise module.

The denoise module has a hidden size of $d' = d/r$, where $r$ is the reduction factor, and has $L$ layers. Each layer $i$ contains a transformer $\mathcal{T}_i$ and a gate vector $\boldsymbol{g}_i$. To integrate the server-side hidden states, the input to layer $i$ is a combination of $H_i$ and the previous layer's output $A_{i-1}$, with the gate vector $\boldsymbol{g}_i$ controlling the proportion of this mixture. The proportion is computed by $\boldsymbol{\mu}_i = \text{sigmoid}(\boldsymbol{g}_i)$. Thus, the input to the transformer $\mathcal{T}_i$ is

$$Z_i = \boldsymbol{\mu}_i A_{i-1} + (1 - \boldsymbol{\mu}_i) H_i^{\text{dn}}, \tag{2}$$

where $H_i^{\text{dn}} \in \mathbb{R}^{n \times d'}$ is the downsampled $H_i$. Specifically, for the first layer $A_{i-1} = E^{\text{dn}}$, $E^{\text{dn}}$ also represents the downsampled $E$. The gating mechanism adjusts the influence of the server-side hidden states on the denoising process, ensuring that the refined hidden state optimally balances the client-side and server-side information.

To further enhance the learning ability of the denoise module, residual connections are introduced, which propagate the information of the initial embeddings to the deeper layers, preserving the integrity of the original signals during denoising. The output of layer $i$ is recursively defined as:

$$A_i = A_{i-1} + \mathcal{T}_i(Z_i). \tag{3}$$

The downsampling process, along with the subsequent upsampling, is learned by linear layers on the server side to reduce communication cost:

$$H_i^{\text{dn}} = \mathcal{W}_i^{\text{dn}}(H_i), \tag{4}$$

$$E^{\text{dn}} = \mathcal{W}_{\text{Emb}}^{\text{dn}}(E). \tag{5}$$

The final output $A_{L-1}$ of the denoising module is then upsampled back to the original dimension $d$ to create the denoised hidden state:

$$H^{\text{denoised}} = \mathcal{W}^{\text{up}}(A_{L-1}). \tag{6}$$

**Optimization** The denoised hidden state is fed into a task-specific head to generate predictions, and the corresponding loss is computed for model optimization. For classification tasks, the head outputs logits, and cross-entropy loss is applied:

$$\hat{y} = W^{\text{task}}(H^{\text{denoised}}), \tag{7}$$

$$\mathcal{L}(\hat{y}, y) = -\sum_i y_i \log(\hat{y}_i), \tag{8}$$

where $y$ represents the vector of ground-truth labels. Both the denoising module and the task-specific parameters are optimized to minimize this loss, improving classification accuracy and denoising effectiveness. This ensures denoised hidden states effectively contribute to the task performance.

## 4.2 COMMUNICATION OVERHEAD REDUCTION

While leveraging all intermediate hidden states yields strong denoising performance, the resulting communication overhead limits practicality. To address this, HiddenEcho incorporates a hidden layer filter and a dimension reducer, effectively balancing model performance with communication efficiency and reducing transmission costs without notable performance loss.

**Hidden Layer Filter** Transmitting all intermediate hidden states between server and client incurs prohibitive communication costs. We observe that not all layers contribute equally to the final output, suggesting that selectively transmitting only the most informative layers could maintain performance while reducing overhead.

To quantify the contribution of each hidden layer to the final output, a gradient-based filter is designed. For a given layer $i (i < L - 1)$, we gradually vary the value of its hidden state from 0 to $H_i$

and observe the corresponding changes in the output of the last layer. Denoting $\mathcal{T}_i^S$ as layer $i$ of the server-side LLM, we have:

$$\hat{H}_{L-1} = \mathcal{T}_{L-1}^S \circ ... \circ \mathcal{T}_i^S(\hat{H}_i), \tag{9}$$

where $\hat{H}_i$ is the current value of layer $i$, and $\hat{H}_{L-1}$ is the output of the last layer corresponding to the hidden state $\hat{H}_i$. $\circ$ signifies the sequential application of layers, with each layer's output feeding into the next layer in the sequence.

The layer's contribution $C_i$ is defined by the cumulative gradient of these output changes:

$$C_i = H_i \int_0^{H_i} \frac{\partial \hat{H}_{L-1}}{\partial \hat{H}_i} d\hat{H}_i. \tag{10}$$

However, in practice, calculating the continuous integral is computationally challenging. Following (Dai et al., 2022), we approximate the integral using Riemann summation with $m$ steps:

$$C_i = \frac{H_i}{m} \sum_{j=1}^{m} \left. \frac{\partial \hat{H}_{L-1}}{\partial \hat{H}_i} \right|_{\hat{H}_i = (j/m)H_i}. \tag{11}$$

This calculation is performed before fine-tuning. A small subset is sampled from the training dataset. Each sample undergoes standard preprocessing: tokenization, embedding, and perturbation, but not denoising. The server computes the layer contributions for each sample using Eq. (11) and averages these contributions across all samples.

Layers with the highest $k$ contributions are selected to minimize communication overhead while maintaining performance, where $k$ is a small hyperparameter. During each forward pass, only these layers' hidden states are transmitted, significantly reducing communication costs. Upon receiving these hidden states, the client's denoising module correspondingly skips unselected layers, accelerating computation and lowering resource requirements.

**Dimension Reducer**  While layer selection reduces the number of transmitted states, each hidden state remains high-dimensional. Projecting the hidden states of the server-side LLM using linear layers is often effective, but it may fail to learn optimal representations due to the lack of explicit optimization objectives. We address this by applying the information bottleneck technique (Alemi et al., 2017) to compress hidden states while preserving task-relevant information.

In `HiddenEcho`, we formulate dimension reduction as an information bottleneck problem: minimize the mutual information (MI) between the noisy embedding $E'$ and the downsampled hidden states $H_i^{\mathrm{dn}}$, while maximizing the MI between the denoised output $H^{\mathrm{denoised}}$ and the downsampled hidden states $H_i^{\mathrm{dn}}$. The corresponding loss function is:

$$\mathcal{L}^{\mathrm{IB}} = \frac{1}{n} \sum_{i=0}^{n-1} I(E'; H_i^{\mathrm{dn}}) - \beta I(H^{\mathrm{denoised}}; H_i^{\mathrm{dn}}). \tag{12}$$

Consequently, the overall model optimization loss is a combination of the task loss and the information bottleneck loss, weighted by $\alpha, \beta$:

$$\mathcal{L} = \mathcal{L}(\hat{y}, y) + \alpha \mathcal{L}^{\mathrm{IB}}. \tag{13}$$

Although exact MI computation for high-dimensional variables is inherently challenging (Belghazi et al., 2018), an exact value is often unnecessary for optimization. Based on this, MINE (Belghazi et al., 2018), a neural network-based approach, is employed to estimate MI effectively. MINE uses a statistics network to learn a function $f_\theta$ that maximizes the difference between its expectation over the joint distribution $P(X, Y)$, and the exponential expectation over the product of the marginal distributions $P(X)P(Y)$. The estimated MI is then approximated by the supremum of this difference. Mathematically, this can be expressed as

$$\max_\theta \left( \mathbb{E}_{P(X,Y)}[f_\theta(X, Y)] - \exp(\mathbb{E}_{P(X)}[\mathbb{E}_{P(Y)}[f_\theta(X, Y)]]) \right). \tag{14}$$

$$I(X; Y) \approx \sup_\theta \left( \mathbb{E}_{P(X,Y)}[f_\theta(X, Y)] - \exp(\mathbb{E}_{P(X)}[\mathbb{E}_{P(Y)}[f_\theta(X, Y)]]) \right). \tag{15}$$

This neural network-based estimator allows for an efficient computation of MI in scenarios where traditional methods are computationally prohibitive.

Specially, we prepare two statistics networks for each hidden state $H_i^{\mathrm{dn}}$: one to estimate the MI $I(E'; H_i^{\mathrm{dn}})$, and the other to estimate $I(H^{\mathrm{denoised}}; H_i^{\mathrm{dn}})$. After calculating the task loss at each step, these statistics networks are optimized for several steps according to Eq. equation 14. Once the optimization process is finished, the networks are used to compute the MI estimates. The information bottleneck loss is computed based on these estimates, as described in Eq. equation 12.

## 5 EXPERIMENTS

We evaluate perturbation methods on text classification and generation tasks using Qwen2-1.5B and Llama3-1B (1.54B and 1.23B parameters) for classification, and T5-Large (0.75B parameters) for generation. Datasets include Financial Phrasebank, MRPC, BBC News, and Tweet Annotation for classification; IWSLT2014, CNN/DailyMail, and Samsum for generation. Details are provided in Appendix J.2. We employ LoRA fine-tuning via Transformers (Wolf et al., 2020) and PEFT (Mangrulkar et al., 2022), with AdamW and a constant scheduler (lr = 1.5e-4). Performance is measured using AUC and Empirical Privacy (Definition 4) for classification (Li et al., 2023a), and BLEU for generation (Papineni et al., 2002). All experiments run on an NVIDIA RTX 3090 GPU.

**Attacks** Following prior studies (Song & Raghunathan, 2020), we evaluate the privacy protection effectiveness of `HiddenEcho` and baseline methods under simulated attacks within the split learning framework (Shen et al., 2023). In our experiments, a white-box attack setting is assumed, where attackers have access to user-submitted text embeddings and the parameters of the embedding model. As described in 3.1, the **Embedding Inversion Attack (EIA)** and **Attribute Inference Attack (AIA)** models are used to evaluate the effectiveness of privacy preservation methods.

Table 1: Performance of different perturbation methods on text classification tasks based on Qwen2-1.5B.

| Dataset | | MRPC | | | | Financial | | | | BBC News | | | |
|---|---|---|---|---|---|---|---|---|---|---|---|---|---|
| Privacy Budget $\eta$ | | 100 | 1000 | 5000 | 6000 | 100 | 1000 | 5000 | 6000 | 100 | 1000 | 5000 | 6000 |
| GAN-DP | AUC | 0.497 | 0.532 | 0.597 | 0.612 | 0.501 | 0.524 | 0.618 | 0.629 | 0.606 | 0.620 | 0.684 | 0.720 |
| | *EP* | *1.000* | *0.999* | *0.999* | *0.998* | *1.000* | *0.999* | *0.997* | *0.992* | *0.995* | *0.991* | *0.971* | *0.962* |
| LDP | AUC | 0.551 | 0.557 | 0.553 | 0.599 | 0.596 | 0.595 | 0.629 | 0.617 | 0.648 | 0.646 | 0.736 | 0.803 |
| | *EP[1]* | *0.988* | *0.987* | *0.956* | *0.867* | *0.988* | *0.987* | *0.967* | *0.886* | *0.973* | *0.972* | *0.914* | *0.820* |
| SnD | AUC | 0.513 | 0.513 | 0.526 | 0.533 | 0.558 | 0.565 | 0.595 | 0.630 | 0.627 | 0.628 | 0.629 | 0.637 |
| `HiddenEcho`-Full | AUC | 0.646 | **0.657** | 0.661 | 0.667 | **0.875** | **0.874** | **0.883** | **0.889** | 0.685 | **0.803** | **0.839** | **0.960** |
| `HiddenEcho` | AUC | **0.660** | 0.655 | **0.666** | **0.668** | 0.857 | 0.855 | 0.860 | 0.866 | **0.732** | 0.747 | 0.805 | 0.951 |
| AUC Improve % | | 19.78 | 15.22 | 11.56 | 9.15 | 46.81 | 46.89 | 40.38 | 41.11 | 12.96 | 24.30 | 13.99 | 19.55 |

[1] The EP of SnD and `HiddenEcho` is consistent with that of LDP, while GAN-DP differs from the other methods. Subsequent tables follow this format in reporting EP.

### 5.1 RESULTS OF EMBEDDING INVERSION ATTACK

We evaluate various methods against embedding inversion attacks in text classification using Qwen2-1.5B under Metric-DP, which is based on $d_\chi$-privacy budgets $\eta = 100, 1000, 5000, 6000$ (definition in Appendix B; results on Llama3-1B are in Appendix J.3). We describe the baseline methods in Appendix J.1. Our proposed approach has two variants: `HiddenEcho`-Full uses all hidden layers for denoising, while `HiddenEcho` selectively transmits high-impact layers via gradient-based filtering to achieve significantly reduced communication. For comparison, we also evaluate SnD, which relies on a fixed pre-trained denoising model.

As shown in Table 1, `HiddenEcho`-Full achieves higher AUC scores, confirming its effectiveness in mitigating noise amplification and delivering the best performance on several datasets, with AUC improvements of up to 46.89% (Financial Phrasebank) and 24.30% (BBC News). Interestingly, the more efficient `HiddenEcho` variant can even outperform HiddenEcho-Full on MRPC (+19.78%) and BBC News (+12.96%), suggesting that not all layers contribute positively to denoising. In contrast, SnD underperforms because its fixed model fails to adapt to the shifting hidden distributions

Table 2: Ablation study of `HiddenEcho` on text classification tasks based on Qwen2-1.5B.

| Dataset | MRPC | | | Financial | | | BBC News | | |
|---|---|---|---|---|---|---|---|---|---|
| Privacy Budget $\eta$ | 100 | 1000 | 5000 | 100 | 1000 | 5000 | 100 | 1000 | 5000 |
| `HiddenEcho` | **0.660** | **0.655** | **0.666** | **0.857** | **0.855** | **0.860** | **0.732** | **0.747** | **0.805** |
| `HiddenEcho` $-Res$ | 0.646 | 0.648 | 0.658 | 0.814 | 0.815 | 0.819 | 0.659 | 0.661 | 0.729 |
| `HiddenEcho` $-HLF$ | 0.637 | 0.640 | 0.641 | 0.773 | 0.773 | 0.774 | 0.629 | 0.630 | 0.719 |
| `HiddenEcho` $-DR$ | 0.632 | 0.649 | 0.644 | 0.789 | 0.799 | 0.801 | 0.630 | 0.663 | 0.789 |

during fine-tuning, leading to ineffective noise removal. See Appendix J.6 for visualization of the baselines' classification. Additional EIA evaluation on text generation is provided in Appendix J.4. We further compare HiddenEcho with the federated learning baseline POPri (Hou et al., 2025) in Appendix J.8.

## 5.2 ABLATION STUDY

We conduct ablation studies on `HiddenEcho`, which subsumes all components. We evaluate three variants: removing residual connections ($-Res$), replacing the Hidden Layer Filter with fixed skip layers ($-HLF$), and substituting the Dimension Reducer with a linear layer ($-DR$). As shown in Table 2, the complete `HiddenEcho` consistently achieves the highest AUC across datasets and privacy budgets. Removing residual connections degrades performance by 1.1%–11.51%, with the largest drop on BBC News (9.4%–11.51%). Replacing the HLF causes the most significant decline—up to 14.1% (e.g., 0.732→0.629 on BBC News at $\eta$=100)—demonstrating the importance of dynamic layer selection in noise suppression. The $-DR$ variant reduces AUC by 0.9%–13.9%, with greater impact on complex tasks (e.g., 6.5%–7.9% drop on Financial).

These results confirm that residual connections stabilize training, the HLF enhances communication and noise control, and the dimension reducer improves feature robustness, collectively ensuring architectural efficacy under DP perturbations.

## 5.3 RESULTS OF ATTRIBUTE INFERENCE ATTACK

Compared to other text classification datasets, the Tweet Annotation dataset includes critical attributes such as the author's age and education, making it well-suited for attribute inference attacks. Following the approach in (Song & Raghunathan, 2020), we train an MLP model to predict related information for each tweet. For detailed architecture, refer to Appendix J.5. Specifically, we evaluate the model's robustness using RMSE for age prediction and Empirical Privacy (EP) for education inference, where higher values indicate stronger resistance to attacks. As depicted in Fig 3, the red dashed line represents the privacy protection capabil-

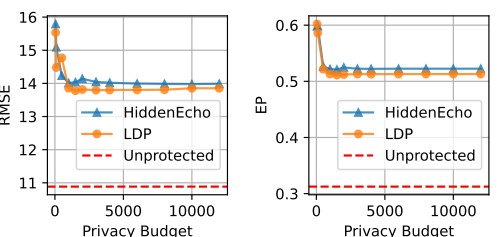

Figure 3: AIA performance on Tweet Annotation Sensitivity 2 (Kern et al., 2023) with Qwen2-1.5B.

ity without perturbation. Both `HiddenEcho` and standard LDP exhibit performance degradation as privacy protection increases. However, except in scenarios with high privacy budgets (e.g., $\eta = 100$), where both methods show nearly comparable, `HiddenEcho` consistently outperforms LDP in terms of privacy protection under other conditions. A discussion of potential privacy risks regarding gradient inversion attacks is provided in Appendix H.

## 5.4 OPTIMIZATION

We report the optimization process of the `HiddenEcho`, with results on the BBC News dataset using Qwen2-1.5B visualized in Fig 4. Specifically, we compare the optimization trajectories of two configurations: `HiddenEcho`-Full, which utilizes full hidden layer states, and `HiddenEcho`, which employs filtered $k$ hidden layers. The evaluation metrics encompass training loss, evaluation

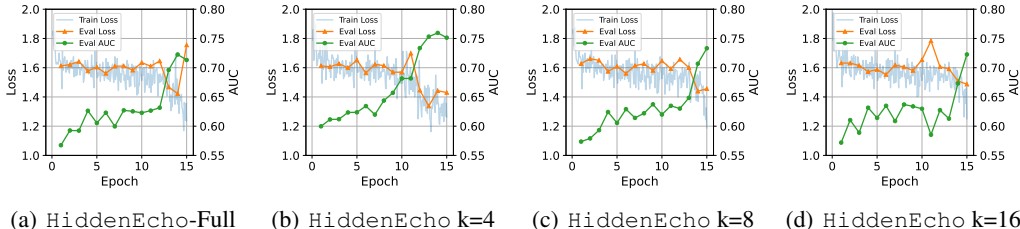

| (a) `HiddenEcho`-Full | (b) `HiddenEcho` k=4 | (c) `HiddenEcho` k=8 | (d) `HiddenEcho` k=16 |

Figure 4: Optmization performance of `HiddenEcho`-Full and `HiddenEcho` with different hidden layers on BBC News based on Qwen2-1.5B.

Table 3: Training time cost overhead of different methods for one epoch (left) and communication cost of `HiddenEcho`-Full (`HE-Full`) and `HiddenEcho` (`HE`) for one batch (right).

| | | Training time cost (Second) | | | | | | Communication cost (MiB) | | |
|---|---|---|---|---|---|---|---|---|---|---|---|
| Approaches | | LDP | GAN-DP | SnD | `HE-Full` | `HE` | Approaches | | `HE-Full` | `HE` | Saved |
| MRPC | Q | 125 | 118 | 248 | 196 | 166 | MRPC | Q | 2.63 | 0.38 | 85.55% |
| Financial | | 74 | 76 | 184 | 115 | 92 | Financial | | 1.97 | 0.28 | 85.79% |
| BBC News | | 95 | 97 | 393 | 118 | 108 | BBC News | | 10.50 | 1.50 | 85.71% |
| IWSLT | T | 25 | 26 | - | 51 | 37 | IWSLT | T | 6.00 | 2.25 | 62.50% |
| CNNDM | | 35 | 37 | - | 64 | 46 | CNNDM | | 8.25 | 3.09 | 62.55% |
| Samsum | | 32 | 33 | - | 62 | 40 | Samsum | | 3.05 | 1.14 | 62.62% |

loss, and evaluation AUC, providing a comprehensive view of model convergence and classification performance. During optimization, `HiddenEcho`-Full shows stable decline in evaluate loss in the early period, while overfitting starting at the 14th epoch, with increased evaluation loss and performance degradation, likely due to the use of full hidden layers for correction. In contrast, we observe the optimization trajectories of `HiddenEcho` with 4, 8, and 16 hidden layers. The 4-layer configuration achieves an AUC above 75% by the 12th epoch. The hidden layer filter in `HiddenEcho` enables more focused corrections, reducing overfitting. These findings suggest that using fewer hidden layers in `HiddenEcho` can lead to faster convergence and lower communication overhead without sacrificing performance.

## 5.5 TIME COST

We compare the time overhead of different methods for perturbing embeddings by recording the training time for one epoch for each method. Statistics are shown in the left side of Table 3, where Q and T denotes Qwen2-1.5B and T5-Large, respectively. Since SnD is not applicable to text generation, we do not report statistics for it in this context. The `HiddenEcho` framework, which builds upon LDP, incurs higher computational overhead compared to LDP alone. However, when compared to SnD, which also includes a denoising module, `HiddenEcho`-Full demonstrates faster training speeds, with time costs reduced by up to 72.52% on the BBC News dataset. Although `HiddenEcho` incorporates additional steps such as a hidden layer filter and dimension reduction, it still achieves faster training speeds due to the use of fewer hidden layers. Notably, while the GAN-DP method based on DNN shows advantages in a single training epoch, it requires a pre-training process for the GAN, which adds to its overall time cost.

## 5.6 COMMUNICATION COST

This section analyzes the communication overhead of `HiddenEcho`. `HiddenEcho` requires transmitting hidden layer states between the server and client to enable correction. The full hidden states are transmitted in `HiddenEcho`-Full, resulting in large data volumes and high real-time transmission demands during LLM fine-tuning. In contrast, `HiddenEcho` compresses communication by selecting key hidden layers for transmission. The communication costs per data batch for both `HiddenEcho` variants are shown in the right side of Table 3. The results indicate that `HiddenEcho` reduces communication overhead by over 60% compared to `HiddenEcho`-Full. Specifically, for text classification tasks, it achieves a remarkable space saving of over 85%. For text

generation tasks, which require `HiddenEcho` to filter more hidden layers to achieve optimal performance, the space saving is approximately 62%. Under typical network bandwidth, client-server communication using `HiddenEcho` remains unaffected. A detailed communication conservation analysis is provided in Appendix E. Additional analysis on inference-time communication overhead is provided in Appendix J.7.

## 6 CONCLUSION

Large language models (LLMs) in the Model-as-a-Service paradigm enable convenient customization but raise privacy concerns. While differential privacy (DP) mitigates these risks, it degrades model performance, especially as injected noise is amplified through multi-layer transformer blocks. To address this, we propose `HiddenEcho`, a split learning-based framework that integrates with hidden layers and supports both fine-tuning and inference. Experiments show that `HiddenEcho` achieves a superior privacy-utility trade-off and significantly improves downstream task performance under DP constraints, offering a novel solution to noise mitigation in privatized LLMs.

### ACKNOWLEDGMENTS

This work was supported in part by Hong Kong RGC Theme-based Research Scheme (TRS) under Grant T43- 513/23-N, in part by the NSFC and Hong Kong RGC Collaborative Research Scheme under Grant 62321166652, in part by the Guangdong Basic and Applied Basic Research Foundation under Grant 2025A1515011996, and in part by the Fundamental Research Funds for the Central Univer- sity under Grant CXTD202406.

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

## A  THE USE OF LARGE LANGUAGE MODELS

The language of this paper was polished using large language models (LLMs) to enhance clarity and readability. The final content and academic integrity remain the responsibility of the authors.

## B  $d_\chi$ PRIVACY

Differential Privacy (DP) is a perturbation-based privacy-preserving mechanism that provides a rigorous framework for safeguarding data confidentiality. By introducing carefully calibrated noise during the training or fine-tuning of LLMs, DP makes it significantly harder to extract sensitive information from the perturbed data (Behnia et al., 2022).

In particular, the $d_\chi$-based Metric-DP method is more suitable for text structural embeddings (Feyisetan et al., 2020). Based on the differential privacy, we define the $d_\chi$-Privacy.

**Definition 3 ($d_\chi$-Privacy)** *Let $X$ be the input domain, $Y$ be the output domain, and $d_\chi$ be a distance metric over $X$. A randomized mechanism $M : X \to Y$ satisfies $\eta d_\chi$-privacy if for any two inputs $x, x' \in X$ and any subset $S \subseteq Y$, the following inequality holds:*

$$\frac{\Pr[M(x) \in S]}{\Pr[M(x') \in S]} \le e^{\eta d_\chi(x,x')}, \tag{16}$$

*where $\eta \ge 0$ represents the privacy budget, controlling the trade-off between privacy and utility.*

`HiddenEcho` offers a novel solution to mitigate LLM performance degradation caused by noise-based differential privacy mechanisms.

## C  PRIVACY DEFINITION

Building on prior research (Coavoux et al., 2018b), which defines privacy as the adversary's inability to infer information about the input from its latent representations, we adopt a similar perspective in our work.

**Definition 4 (Empirical Privacy)** *Empirical Privacy (EP) quantifies the adversary's inability to reconstruct the original input or infer sensitive attributes from perturbed text. The degree of privacy protection increases as it becomes more challenging for an attacker to recover the original text or extract sensitive information.*

$$EP = 1 - \frac{\sum_{x_i \in X} \mathbb{I}(f(\Phi(x_i)), x_i)}{|X|}, \tag{17}$$

*where $\Phi(x_i)$ represents the embedding layer of the LLM, $f$ denotes a general inversion process, and $\mathbb{I}$ indicates the correct predictions.*

## D  TIME AND SPACE COMPLEXITY

### D.1  HIDDENECHO-FULL

The computational cost of `HiddenEcho`-Full is primarily driven by its denoising module. For the time complexity:

1. Transformer Layers: Each Transformer layer processes hidden states with a complexity of $O(n^2 d' + nd'^2)$, where $d' = d/r$ (reduced hidden size), $n$ is the sequence length, and $L$ is the number of layers. The total complexity for all layers is:

$$O(L(n^2 d/r + nd^2/r^2)).$$

2. Down/Upsampling: The linear transformations for downsampling and upsampling the embeddings have a complexity of $O(Lndd')$.

3. Computing gate vectors and performing mixing operations incurs a complexity of $O(Lnd')$.

Combining these, the total time complexity is:

$$O(L(n^2d/r + nd^2/r^2 + nd^2/r)).$$

For the space complexity:

1. Parameter storage: The Transformer layers and linear transformations require $O(Ld'^2 + Ldd')$ for storing parameters.

2. Intermediate Representations: The hidden states and gate vectors contribute $O(Lnd' + Ld')$ to memory usage.

Thus, the total space complexity is:

$$O(L(d^2/r^2 + nd/r + d^2/r)).$$

## D.2   HIDDENECHO

To address the high communication overhead, `HiddenEcho` compresses the hidden layer states using selective filtering and dimensionality reduction. For the time complexity:

1. Hidden Layer Filter: Estimating the gradient $\frac{\partial \hat{H}_{L-1}}{\partial \hat{H}_i}$ for each approximation step involves backpropagation through the layers following $H_i$. This incurs a complexity of $O(mn^2d)$ per layer, where $m$ denotes the number of approximation steps. Summing across $L$ layers, the total cost is:
$$O(mLn^2d).$$

2. Dimension Reducer: Downsampling and upsampling hidden states incur $O(ndd')$, where $d' = d/r$ is the reduced dimension, and $r$ is the reduction factor. MINE operations over $n_H$ selected layers require $O(knn_Hd')$, where $k$ is the optimization steps for MINE.

The total time complexity is:

$$O(mLn^2d + knn_Hd/r + nd^2/r).$$

For the space complexity:

1. Hidden Layer Filter: Requires $O(Lnd)$ for storing gradients and contributions.

2. Dimension Reducer:   MINE statistics networks require $O(n_Hd'^2)$.   Downsampled/upsampled states add $O(nn_Hd')$.

The total space complexity is:

$$O(Lnd + n_Hd^2/r^2 + nn_Hd/r).$$

## E   COMMUNICATION ANALYSIS

In the `HiddenEcho`-Full, all $L$ hidden states of the server-side LLM are transmitted. Each hidden state has dimensions of $n \cdot d$, where $n$ represents the sequence length and $d' = d/r$ denotes the reduced hidden dimension achieved via dimensionality reduction by a factor $r$. The total communication volume can be expressed as:

$$V_{\texttt{HiddenEcho-Full}} = L \cdot n \cdot d'.$$

In contrast, the `HiddenEcho` configuration transmits only $n_H$ selected hidden layers, resulting in a total communication volume of:

$$V_{\texttt{HiddenEcho}} = n_H \cdot n \cdot d'.$$

To quantify the reduction in transmission, the ratio of communication volumes between the two configurations is given by:

$$\frac{V_{\texttt{HiddenEcho}}}{V_{\texttt{HiddenEcho}\text{-Full}}} = \frac{n_H \cdot n \cdot d'}{L \cdot n \cdot d'} = \frac{n_H}{L}.$$

The percentage of transmission volume saved is therefore:

$$\text{Savings (\%)} = \left(1 - \frac{n_H}{L}\right) \cdot 100.$$

**Example Case**: When $n_H \ll L$, significant communication savings can be achieved. For instance, consider $n_H = 4$ and $L = 28$. The percentage savings in transmission volume is calculated as:

$$\text{Savings (\%)} = \left(1 - \frac{4}{28}\right) \cdot 100 \approx 87.50\%.$$

## F  PROOF OF NOISE MITIGATION IN HIDDENECHO

We provide proof demonstrating how the `HiddenEcho`-Full framework mitigates interlayer noise amplification by analyzing noise propagation through transformer layers and the corrective effects of the denoising module.

### F.1  NOISE AMPLIFICATION IN TRANSFORMER LAYERS

Let the hidden state at the $i$-th layer be $H_i$, and the corresponding noise be $\delta_i$. The hidden state at the $(i + 1)$-th layer can be expressed as:

$$H_{i+1} = \mathcal{T}_{i+1}(H_i + \delta_i),$$

where $\mathcal{T}_{i+1}$ represents the transformer operation. Due to the nonlinear nature of $\mathcal{T}_{i+1}$, noise $\delta_i$ propagates and is amplified. The noise at the $(i + 1)$-th layer can be approximated as:

$$\delta_{i+1} = f(\delta_i),$$

where $f(\cdot)$ denotes the transformation applied by the layer. The magnitude of $\delta_{i+1}$ is bounded by the Jacobian norm of the transformation:

$$\|\delta_{i+1}\| \leq \|J_f(H_i)\| \cdot \|\delta_i\|,$$

where $\|J_f(H_i)\|$ is the Jacobian norm. Defining the noise amplification factor as $\alpha_i = \mathbb{E}[\|J_f(H_i)\|]$, we obtain:

$$\|\delta_{i+1}\| \leq \alpha_i \|\delta_i\|, \quad \text{where } \alpha_i > 1.$$

Over $L$ layers, the noise at the final layer is amplified as:

$$\|\delta_L\| \leq \prod_{i=1}^{L} \alpha_i \|\delta_0\|,$$

where $\delta_0$ denotes the initial noise introduced by the privacy-preserving mechanism.

### F.2  NOISE DECOMPOSITION AND DENOISING

The hidden state $H_i$ can be decomposed into two components:

$$H_i = S_i + \delta_i,$$

where:

- $S_i$: Signal component containing task-relevant information.
- $\delta_i$: Noise component introduced for privacy preservation.

The `HiddenEcho` module $\mathcal{D}$ utilizes the noise-free initial embedding $E$ and the set of server-side hidden states $\boldsymbol{H} = \{H_0, H_1, \ldots, H_{L-1}\}$ to produce a denoised hidden state:
$$H_i^{\text{denoised}} = \mathcal{D}(E, \boldsymbol{H}).$$

The denoised hidden state can be expressed as:
$$H_i^{\text{denoised}} = S_i + \delta_i^{\text{denoised}},$$
where $\delta_i^{\text{denoised}}$ represents the residual noise after applying the denoising module.

### F.3 DYNAMIC MIXING AND RESIDUAL CONNECTIONS

The `HiddenEcho` module incorporates dynamic mixing and residual connections to enhance signal retention and suppress noise. The input to the $i$-th layer of the module is given by:
$$Z_i = \mu_i A_{i-1} + (1 - \mu_i) H_i^{\text{dn}},$$
where:

- $A_{i-1}$: Output from the previous layer with reduced noise.
- $H_i^{\text{dn}} = \mathcal{W}^{\text{dn}}(H_i)$: Compressed version of the hidden state, containing both signal and noise.

The gate parameter $\mu_i \in (0, 1)$ dynamically adjusts the contributions of $A_{i-1}$ and $H_i^{\text{dn}}$. Expanding $Z_i$ in terms of its components:
$$Z_i = \mu_i(S_{A_{i-1}} + \delta_{A_{i-1}}) + (1 - \mu_i)(S_{H_i} + \delta_{H_i}).$$

The contributions of signal and noise can be written as
$$S_{Z_i} = \mu_i S_{A_{i-1}} + (1 - \mu_i) S_{H_i}, \quad \delta_{Z_i} = \mu_i \delta_{A_{i-1}} + (1 - \mu_i) \delta_{H_i}.$$

Using the triangle inequality, the noise magnitude satisfies:
$$\|\delta_{Z_i}\| \leq \mu_i \|\delta_{A_{i-1}}\| + (1 - \mu_i)\|\delta_{H_i}\|.$$
This demonstrates the effectiveness of dynamic mixing and residual connections in amplifying the signal while suppressing sparse noise. Generally, it ensures that $\|\mathcal{D}(\delta, E, H)\| > 0$.

### F.4 NOISE REDUCTION AT THE FINAL LAYER

The residual noise after denoising is given by:
$$\|\delta^{\text{denoised}}\| = \|\delta\| \cdot \left(1 - \frac{\|\mathcal{D}(\delta, E, H)\|}{\|\delta\|}\right).$$

We have $\|\mathcal{D}(\delta, E, H)\| > 0$, ensuring:
$$0 < 1 - \frac{\|\mathcal{D}(\delta, E, H)\|}{\|\delta\|} < 1,$$

which implies:
$$\|\delta^{\text{denoised}}\| < \|\delta\|.$$

Let $0 < \beta = \frac{\|\delta^{\text{denoised}}\|}{\|\delta\|} < 1$. The corrected noise at the $i$-th layer satisfies:
$$\|\delta_i^{\text{denoised}}\| \leq \beta_i \|\delta_i\|.$$

At the $(i + 1)$-th layer, the noise satisfies:
$$\|\delta_{i+1}^{\text{denoised}}\| \leq \beta_{i+1} \alpha_i \|\delta_i^{\text{denoised}}\|.$$

By recursively applying this relationship across $L$ layers, the noise at the final layer satisfies:
$$\|\delta_L^{\text{denoised}}\| \leq \left(\prod_{i=1}^{L} \beta_i \alpha_i\right) \|\delta_0\| < \prod_{i=1}^{L} \alpha_i \|\delta_0\| = \|\delta_L\|.$$

# G    COMPARING WITH DP

## G.1    PRIVACY GUARANTEE UNDER EMBEDDING-BASED INVERSION

We first analyze the privacy strength of `HiddenEcho` compared with the standard DP mechanism under Embedding-based Inversion. Let the clean embedding be $E$ and the added DP noise be $\delta$, such that the privatized embedding is

$$E' = E + \delta.$$

Since the randomization is fully applied at the client side, the transmitted $E'$ already satisfies the DP constraint with privacy budget $\eta$. By the *post-processing property* of differential privacy, any further mapping of $E'$ (e.g., the server computing hidden states $H = B(E')$ and returning them to the client) does not weaken the privacy guarantee. Therefore, the overall mechanism of `HiddenEcho` satisfies the same $\eta$-DP guarantee as the baseline DP approach:

$$\text{DP\_budget}_{\texttt{HiddenEcho}} = \text{DP\_budget}_{\text{DP}} = \eta.$$

## G.2    MODEL ACCURACY AND NOISE AMPLIFICATION

Next, we compare robustness to noise amplification across transformer layers. Denote by $\alpha_i$ the amplification factor of the $i$-th layer. For the baseline DP mechanism, the accumulated noise at the final layer $L$ is bounded by

$$\|\delta_L\| \leq \Big(\prod_{i=1}^{L} \alpha_i\Big) \|\delta_0\|,$$

where $\delta_0$ is the initial DP noise at the embedding layer. In `HiddenEcho`, a lightweight client-side correction is applied at each layer with suppression factor $\beta_i \in (0,1)$, yielding

$$\|\delta_L^{\text{den}}\| \leq \Big(\prod_{i=1}^{L} \beta_i \alpha_i\Big) \|\delta_0\|.$$

Since $\beta_i < 1$, we have

$$\|\delta_L^{\text{den}}\| < \|\delta_L\|,$$

which shows that `HiddenEcho` effectively suppresses inter-layer noise amplification and preserves task-relevant signals under the same DP budget.

## G.3    COMMUNICATION COST

Finally, we compare the communication overhead. For the baseline DP mechanism, transmitting only embeddings requires

$$V_{\text{DP}} = n \cdot d,$$

where $n$ is the sequence length and $d$ the embedding dimension. For `HiddenEcho`-Full, all $L$ hidden layers are downsampled to dimension $d' = d/r$, resulting in

$$V_{\texttt{HiddenEcho-full}} = L \cdot n \cdot d'.$$

In the communication-efficient variant `HiddenEcho`, only $n_H$ critical layers are transmitted, giving

$$V_{\texttt{HiddenEcho}} = n_H \cdot n \cdot d'.$$

The relative saving ratio is

$$\text{Savings} = 1 - \frac{V_{\texttt{HiddenEcho}}}{V_{\texttt{HiddenEcho-Full}}} = 1 - \frac{n_H}{L}.$$

For example, if $L = 28$ and $n_H = 4$, the saving is $87.5\%$, which aligns with our experimental results showing more than $85\%$ reduction in classification tasks.

# H  POTENTIAL PRIVACY RISKS

HiddenEcho's denoising procedure builds on a one-shot Local Differential Privacy (LDP) perturbation of client-side embeddings. Consequently, against embedding-based inversion attacks, HiddenEcho inherits the formal privacy guarantees of LDP: since the server receives only the perturbed embedding (E' = E + $\delta$), all subsequent processing is protected by DP post-processing invariance.

A different situation arises under gradient-based reconstruction attacks, because HiddenEcho requires returning certain gradient signals from the client-side denoiser to the server during training. Under our threat model, an eavesdropping adversary may intercept these gradients. In such cases, HiddenEcho no longer benefits from a provable DP guarantee, since the gradient may, in principle, encode additional information about the client input.

However, mainstream gradient inversion techniques (e.g., Deep Leakage from Gradients (Zhu et al., 2019) and follow-up work) rely fundamentally on a white-box optimization pipeline: they iteratively search for a "virtual input" whose gradients are computed using the known model architecture and parameters and match the intercepted gradients. White-box access (or a surrogate with high structural fidelity) is crucial for high-quality recovery.

Under HiddenEcho's deployment setting, the adversary does not have access to the server-side LLM parameters or weights. They observe only (i) perturbed embeddings and (ii) a small subset of gradient signals from the denoiser. As summarized in the recent survey of Zhang et al. (2023a), when model parameters are unavailable, gradient inversion becomes dramatically harder: attackers require additional priors, surrogate models, or complex meta-optimization, and recovery quality degrades substantially. Black-box/gray-box scenarios are far less effective than white-box settings.

Thus, while HiddenEcho does not offer a formal DP guarantee under gradient interception, the practical feasibility of such reconstruction attacks is significantly constrained by the absence of model parameters.

# I  WORKFLOW OF HIDDENECHO

Algorithm 1 outlines the training process for `HiddenEcho`.

---

**Algorithm 1** Workflow of a Training Step of `HiddenEcho`

---

**Require:** Input tokens $x$, grouth truth $y$
**Ensure:** Loss
    **Client Phase**
  1: Embed tokens: $E \leftarrow \mathcal{E}(x)$;
  2: Inject sampled noise to $E$: $E' \leftarrow E + \delta$;
  3: Send $E'$ to server;
    **Server Phase**
  4: Compute hidden states: $\boldsymbol{H} \leftarrow \mathcal{B}(E')$;
  5: Filter the hidden states according to the precomputed layer contributions to create a subset $\boldsymbol{S}$;
  6: Downsample the hidden states in $\boldsymbol{S}$ by Eq. equation 4;
  7: Return the downsampled $\boldsymbol{S}$ to client;
    **Client Phase**
  8: Compute downsampled embeddings $E^{\mathrm{dn}}$ by Eq. equation 5;
  9: Denoising: $H_{\mathrm{denoised}} \leftarrow \mathcal{D}(E^{\mathrm{dn}}, \boldsymbol{S})$;
10: Compute task loss $\mathcal{L}_{\mathrm{task}}$ by Eq. equation 7 and Eq. equation 8;
11: Optimize the MI estimators by Eq. equation 14;
12: Compute information bottleneck loss $\mathcal{L}_{\mathrm{IB}}$ by Eq. equation 12;
13: Compute total loss $\mathcal{L}$ by Eq. equation 13;
14: **return** Loss $\mathcal{L}$;

---

## J Experimental Supplements

### J.1 Baselines

We evaluate `HiddenEcho` against several strong baselines within the segmented framework, encompassing standard DP algorithms, DP-based denoising methods, and DNN-based perturbation approaches. The baselines include:

- Local Differential Privacy (LDP): Embeddings fed into the LLM's word embedding layer are perturbed with $d_\chi$-noise (Qu et al., 2021), then transmitted to the server.

  In the standard LDP framework for language model inference, the client first maps each input token $x$ to its corresponding dense embedding $e = \text{Embed}(x) \in \mathbb{R}^d$. To satisfy $\epsilon$-local differential privacy under the metric $d_\chi(e, e') = \|e - e'\|_2$, the client adds calibrated noise $\eta$ drawn from the multivariate Laplace mechanism:

  $$\tilde{e} = e + \eta, \quad \text{where} \quad p(\eta) = \frac{\epsilon^d}{C_d B^d} \exp\left(-\frac{\epsilon\|\eta\|_2}{B}\right), \tag{18}$$

  with $B = \sup_{x \sim x'} \|\text{Embed}(x) - \text{Embed}(x')\|_2$ denoting the $L_2$ sensitivity of the embedding function, and $C_d = 2^{d/2}\pi^{d/2}\Gamma(d/2)$ being the surface area of the unit sphere in $\mathbb{R}^d$. This distribution ensures that for any neighboring inputs $x$ and $x'$, the resulting perturbed embeddings satisfy

  $$\frac{p(\tilde{e} \mid x)}{p(\tilde{e} \mid x')} \leq \exp(\epsilon), \tag{19}$$

  which is the formal guarantee of $\epsilon$-$d_\chi$-privacy. The privatized embedding $\tilde{e}$ is then transmitted to the server, which performs downstream inference using the standard LLM architecture without any modification.

- GAN-DP: A GAN-based noise addition method designed to perturb embeddings by introducing $d_\chi$-based noise of varying magnitudes to generate perturbed vectors.

  In the GAN-DP, a generative adversarial network synthesizes privacy-preserving noise adapted to the geometry of the embedding space under the $d_\chi$-privacy notion, where $d_\chi(e, e') = \|e - e'\|_2$. The generator $G_\phi$ learns to produce adaptive noise vectors conditioned on the clean embedding $e$ and a target privacy budget $\epsilon$, while the discriminator $D_\psi$ distinguishes between natural (unperturbed) and perturbed embeddings to preserve semantic utility. Given an input token embedding $e = \text{Embed}(x) \in \mathbb{R}^d$, the client samples a latent vector $z \sim \mathcal{N}(0, I)$ and generates privacy-aware noise as

  $$\eta = G_\phi(e, z; \epsilon), \tag{20}$$

  which is added to the original embedding to yield the privatized representation

  $$\tilde{e} = e + \eta. \tag{21}$$

  During training, the generator is optimized such that the induced distribution over $\tilde{e}$ approximates the exponential mechanism required for $\epsilon$-$d_\chi$-privacy:

  $$p(\tilde{e} \mid e) \propto \exp\left(-\epsilon \cdot \|\tilde{e} - e\|_2 / B\right),$$

  where $B = \sup_{x \sim x'} \|\text{Embed}(x) - \text{Embed}(x')\|_2$ denotes the $L_2$ sensitivity. The adversarial objective further encourages $\tilde{e}$ to remain close to the manifold of real embeddings, balancing privacy and utility. Once trained, only the generator $G_\phi$ is deployed on the client side, enabling efficient, on-device generation of privacy-compliant embeddings without server interaction during inference.

- SnD (Mai et al., 2024): A DP-based denoising approach where the denoising module is pre-trained on the server and then downloaded to the client for noise correction.

  In the SnD framework, the client first computes a token embedding $e = \text{Embed}(x)$ from the private input $x$, then perturbs it with calibrated noise to satisfy $\epsilon$-$d_\chi$-privacy under the $L_2$ metric, yielding the privatized embedding

  $$\tilde{e} = e + \eta, \quad \text{with} \quad p(\eta) \propto \exp\left(-\frac{\epsilon\|\eta\|_2}{B}\right), \tag{22}$$

Table 4: Performance of different perturbation methods on text classification tasks based on Llama3-1B.

| Dataset | | MRPC | | | Financial | | | BBC News | | |
|---|---|---|---|---|---|---|---|---|---|---|
| Privacy Budget $\eta$ | | 1000 | 4000 | 5000 | 1000 | 4000 | 5000 | 1000 | 4000 | 5000 |
| GAN-DP | AUC | 0.506 | 0.502 | 0.513 | 0.540 | 0.550 | 0.576 | 0.619 | 0.647 | 0.664 |
| | *EP* | *0.999* | *0.998* | *0.998* | *0.999* | *0.999* | *0.997* | *0.999* | *0.989* | *0.986* |
| LDP | AUC | 0.489 | 0.529 | 0.494 | 0.561 | 0.567 | 0.559 | 0.619 | 0.627 | 0.641 |
| | *EP* | *0.951* | *0.889* | *0.809* | *0.952* | *0.897* | *0.848* | *0.903* | *0.803* | *0.700* |
| SnD | AUC | 0.509 | 0.504 | 0.507 | 0.558 | 0.553 | 0.572 | 0.632 | 0.633 | 0.633 |
| `HiddenEcho`-Full | AUC | **0.654** | **0.659** | **0.663** | **0.894** | **0.906** | **0.905** | **0.978** | **0.978** | **0.978** |
| `HiddenEcho` | AUC | 0.645 | 0.653 | 0.655 | 0.828 | 0.824 | 0.829 | 0.971 | 0.972 | 0.974 |
| AUC Improve % | | 28.48 | 24.57 | 29.24 | 59.36 | 59.79 | 57.12 | 54.75 | 51.16 | 47.29 |

Table 5: Statistics of datasets.

| Dataset | Task | #Train | #Dev | #Test |
|---|---|---|---|---|
| FP | sentiment analysis | 1,811 | 226 | 227 |
| MRPC | semantic equivalence judgment | 3,301 | 1,725 | 1,725 |
| BBC News | news topic classification | 1225 | 500 | 500 |
| Tweet | offensive speech detection | 1500 | 500 | 500 |
| IWSLT | machine translation | 1,044 | 130 | 131 |
| CNNDM | summarization | 1,322 | 50 | 47 |
| Samsum | summarization | 2,916 | 171 | 150 |

where $B$ is the $L_2$ sensitivity of the embedding function. The client sends $\tilde{e}$ to the server, which performs the main LLM inference:

$$y = \text{LLM}_{\text{server}}(\tilde{e}). \tag{23}$$

The server returns $y$ to the client, who then applies a pre-trained denoising module $D_\theta$—downloaded from the server and trained on public data with synthetic $d_\chi$-compliant noise—to refine the result using knowledge of the original input $x$ and the privacy parameters $(\epsilon, B)$:

$$\hat{y} = D_\theta(y; x, \epsilon, B). \tag{24}$$

This client-side denoising step mitigates utility degradation caused by privacy-preserving perturbation while preserving the formal $\epsilon$-$d_\chi$-privacy guarantee of the initial encoding.

- `HiddenEcho`-Full: Our end-to-end client-side denoising method transmits the full LLM hidden states for processing.

- `HiddenEcho`: Featuring gradient-based hidden layer filtering and dimensionality reduction via information bottleneck theory to lower communication overhead while preserving performance.

### J.2 DATASET DETAILS AND BASE PERFORMANCE

For the text classification task, we utilize:

- Financial Phrasebank (Malo et al., 2014): A sentiment classification dataset with 4,840 financial news sentences, categorized by annotator agreement rates.
- Microsoft Research Paraphrase Corpus (Wang et al., 2018): A sentence pairs dataset collected from news articles, each labeled by human annotators to indicate whether the pairs are paraphrases.

Table 6: Performances of centralized fine-tuning on six datasets for each LLMs.

| Text Classification | | | | |
|---|---|---|---|---|
| Base Model | Metric | MRPC | Financial | BBC News |
| Qwen2-1.5B | AUC | 0.920 | 0.976 | 0.998 |
| Llama3-1B | AUC | 0.928 | 0.980 | 0.999 |
| Text Generation | | | | |
| Base Model | Metric | IWSLT | CNNDM | Samsum |
| T5-large | BLEU | 34.047 | 17.738 | 24.371 |

Table 7: Performance of different perturbation methods on text generation tasks based on T5-Large.

| Dataset | | IWSLT | | | CNNDM | | | Samsum | | |
|---|---|---|---|---|---|---|---|---|---|---|
| Privacy Budget $\eta$ | | 20 | 30 | 40 | 20 | 30 | 40 | 20 | 30 | 40 |
| GAN-DP | BLEU | 0.109 | 10.309 | **29.816** | **5.461** | **13.572** | 12.697 | 4.120 | 4.964 | 5.509 |
| | EP | *0.883* | *0.821* | *0.799* | *0.460* | *0.372* | *0.348* | *0.503* | *0.461* | *0.449* |
| LDP | BLEU | 0.035 | 15.553 | 24.576 | 0.764 | 7.974 | 12.107 | 2.403 | 14.602 | 20.235 |
| | EP | *0.994* | *0.970* | *0.914* | *0.987* | *0.916* | *0.764* | *0.989* | *0.931* | *0.806* |
| HiddenEcho-Full | BLEU | **1.092** | 20.080 | 26.366 | 2.915 | 11.617 | 12.323 | **4.618** | **20.636** | **21.851** |
| HiddenEcho | BLEU | 0.824 | **22.403** | 25.654 | 0.971 | 10.925 | **12.718** | 4.323 | 18.192 | 20.867 |

- BBC News (Greene & Cunningham, 2006): Consists of articles published on the BBC News between 2004 and 2005, with each article categorized into one of five topics: business, entertainment, politics, sports, or technology.
- Tweet Annotation (Kern et al., 2023): A dataset comprises annotated tweet data for hate speech and offensive language under five experimental conditions, which are utilized for attribute inference attacks.

For the text generation task, we utilize:

- IWSLT2014 (IWSLT) (201, 2014): A dataset for English-to-French machine translation, focusing on spoken language.
- CNN DailyMail Short (CNNDM) (Nallapati et al., 2016): A concise version of CNN DailyMail news summaries, paired with fill-in-the-blank questions.
- Samsum Short (Samsum): A shortened version from (Gliwa et al., 2019), comprising messenger-style dialogues with corresponding summaries.

More dataset statistics are reported in Table 5. For reference, the ground truth performance of each large model across various datasets is provided in Table 6.

## J.3 EIA AGAINST FOR TEXT CLASSIFICATION BASED ON LLAMA3-1B

Furthermore, we extend to evaluate the performance of baselines against EIA in text classification tasks using Llama3-1B. Given the significant differences in embedding layer parameter scales across different LLMs, privacy budgets of 1000, 4000, and 5000 are selected for this experiment. All other experimental settings are consistent with those outlined in 5.1. The detailed results are presented in Table 4.

In contrast to Qwen2-1.5B, HiddenEcho-Full exhibits clear superiority when applied to Llama3, achieving significantly higher improvements over baselines, with a maximum performance gain of

59.79%. Although `HiddenEcho` typically performs slightly below `HiddenEcho`-Full, it remains a more advantageous choice in bandwidth-constrained scenarios.

### J.4 EIA AGAINST FOR TEXT GENERATION BASED ON T5-LARGE

We evaluate machine translation on the IWSLT dataset and text summarization on the CNN DailyMail Short and Samsum Short datasets, using T5-Large as the base model. The BLEU scores of `HiddenEcho` and other baseline methods are assessed against EIA at varying $\eta$. Note that SnD's noise reduction model, which processes classification vectors, is unsuitable for text generation tasks.

As shown in Table 7, `HiddenEcho`-Full consistently demonstrates near-optimal performance. On the IWSLT dataset, `HiddenEcho`-Full achieves the highest BLEU scores at $\eta = 20$ (1.092) and $\eta = 40$ (26.366), while `HiddenEcho` outperforms at $\eta = 30$ (22.403). A similar trend is observed on the CNNDM dataset, although `HiddenEcho`-Full performs suboptimally at lower privacy budgets.

The Samsum dataset further confirms `HiddenEcho`-Full's effectiveness, with `HiddenEcho`-Full consistently delivering the highest BLEU scores across all privacy budgets (4.618 at $\eta = 20$, 20.636 at $\eta = 30$, and 21.851 at $\eta = 40$). `HiddenEcho`-Full significantly outperforms GAN-DP and LDP, particularly at lower privacy budgets.

`HiddenEcho`-Full strikes a better balance between privacy and utility in text generation, maintaining competitive EP values while achieving significantly higher generation quality, particularly in summarization tasks.

### J.5 AIA MODEL ARCHITECTURE

The architecture of the attacker model for attribute inference attacks is detailed in Table 8. The model's output size is set to 4 for education inference and 1 for age prediction.

Table 8: Attacker Model Architecture

| Layer | Shape |
|-------|-------|
| Input | Batch size $\times$ 1536 |
| FC | $1536 \times 768$ |
| ReLU | - |
| FC | $768 \times$ Output size |

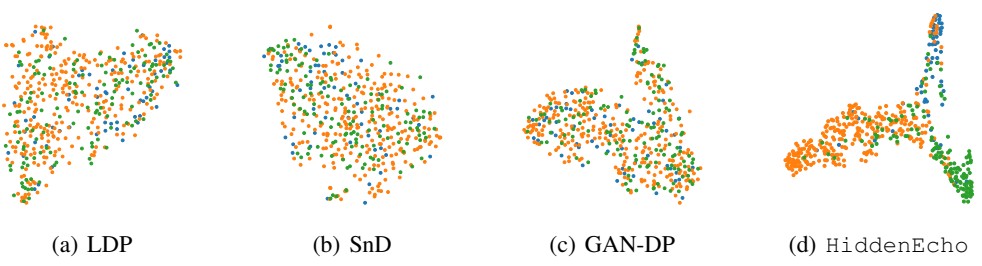

| (a) LDP | (b) SnD | (c) GAN-DP | (d) `HiddenEcho` |

Figure 5: Comparison of visualization of t-SNE between baselines and `HiddenEcho` on the Financial Phrasebank with Qwen2-1.5B.

### J.6 VISUALIZATION

Additionally, we extract the output of the final layer of the server-side LLM after training convergence and employ t-SNE (Van der Maaten & Hinton, 2008) to project the embeddings into a 2D space, maintaining consistent settings across all methods. This visualization enables a comparative

analysis of the effects of different perturbation techniques on the feature space. Each perturbation algorithm is evaluated under the same privacy budget $\epsilon$.

We conduct experiments using four perturbation baselines on the Financial Phrasebank dataset with the Qwen2-1.5B model and $\epsilon$ of 5000. The results are visualized in Fig 5.

The visualization of `HiddenEcho` reveals a triangular spatial distribution of clusters, with points from the same category forming compact groups. This clustering pattern is especially evident in the orange and green categories, highlighting effective feature separation. In contrast, other methods fail to form distinct clusters, with nodes exhibiting dispersed and overlapping distributions. The lack of clear intra-class cohesion and inter-class separation in the embedding space leads to their suboptimal performance.

## J.7 ADDITIONAL COMMUNICATION COST

To further demonstrate the practical efficiency of HiddenEcho, we also measured its inference-time communication overhead on three representative generation benchmarks: IWSLT (machine translation), CNNDM (abstractive summarization), and SamSum (dialogue summarization). As shown in Table 9, the reported values represent the per-sample autoregressive communication cost (normalized relative to full activation transmission), where Avg denotes the mean across all samples in the dataset, and Min/Max indicate the smallest and largest costs observed for any single sample.

Notably, even on long-output tasks like CNNDM, the average overhead remains below 0.75×, with many samples (e.g., in SamSum) requiring as little as 0.17×. This confirms that HiddenEcho effectively reduces communication during decoding, especially by leveraging incremental updates and state caching between Prefill and Decode stages

Table 9: Inference communication cost overhead of `HiddenEcho` for one epoch.

| Dataset | Avg | Min | Max |
|---|---|---|---|
| IWSLT | 0.27MiB | 0.12MiB | 0.92MiB |
| CNNDM | 0.73MiB | 0.30MiB | 1.09MiB |
| Samsum | 0.28MiB | 0.17MiB | 0.44MiB |

## J.8 COMPARISON WITH FEDERATED LEARNING METHOD

Table 10: Performances of POPri on different tasks based on Qwen2-1.5B.

| Task | Metric | Task | Metric |
|---|---|---|---|
| Classification | AUC | Generation | BLEU |
| Financial | 0.615 | IWSLT | 30.604 |
| MRPC | 0.596 | CNNDM | 10.570 |
| BBC News | 0.727 | Samsum | 8.231 |

To enable a more comprehensive comparison with state-of-the-art federated learning approaches, we report the performances of POPri Hou et al. (2025) on different tasks in Table 10. It is important to note that direct alignment of privacy settings between our method and POPri is not feasible, as their privacy-preserving mechanism fundamentally differs from ours. POPri leverages synthetic data generation optimized via client DP feedback, while our approach relies on split learning with client-side noise injection. To ensure fair comparison, we adopt the original privacy parameters reported in POPri's experiments without modification.

The results show that POPri generally underperforms on text classification tasks. It achieves relatively better performance only on the IWSLT translation task in terms of BLEU score, but still lags behind our method on most other benchmarks.

## J.9 PARAMETER SENSITIVITY OF INFORMATION BOTTLENECK

Our experiments on the Financial dataset (Table 11) show that the hyperparameter $\beta$ critically balances privacy and utility: at $\beta = 0.1$, AUC is only 0.779 due to insufficient task-relevant signal retention; performance peaks at $\beta = 0.5$ (AUC = 0.826) and $\beta = 1$ (AUC = 0.823); but rises sharply to 0.612 when $\beta = 5$. This is because $\beta$ governs a trade-off, which targets on minimizing mutual information between noisy embeddings and compressed states (for privacy) while preserving mutual information between denoised outputs and compressed states (for utility). Values of $\beta \in [0.5, 1]$ achieve the optimal balance between these competing objectives.

Table 11: Sensitivity of $\beta$ on Finantial dataset based on Qwen2-7B.

| $\beta$ | 0.1 | 0.5 | 1 | 5 |
|---|---|---|---|---|
| AUC | 0.779 | 0.826 | 0.823 | 0.612 |

## J.10 ADDITIONAL ABLATION RESULTS

Table 12: Additional Ablation Study of `HiddenEcho` on Financial dataset based on Qwen2-7B.

| Privacy Budget | 100 | 1000 | 5000 | 6000 |
|---|---|---|---|---|
| A | 0.573 | 0.571 | 0.576 | 0.577 |
| B | 0.567 | 0.569 | 0.565 | 0.569 |

We added two comparison schemes for the ablation studies: Scheme A uses the original denoising module structure, with inputs limited to dimension-reduced clean embeddings and the final-layer hidden state (in the split learning architecture, the server only feeds back the final hidden layer to the client to complete the task prediction loop, which is an inherent constraint of data interaction under this paradigm, so the last hidden layer must serve as the fixed input benchmark); Scheme B takes noisy embeddings and intermediate hidden states as denoising module inputs. Scheme A isolates the independent contribution of clean embeddings in denoising, while Scheme B highlights the role of the server's intermediate hidden layers in noise correction.

Experiments on the FP dataset and Qwen2-1.5B model show both schemes performed poorly. This confirms dimensionality-reduced clean embeddings and server-side noisy hidden states are complementary and indispensable. The former provides the basic noise correction signal, while the latter delivers task-related deep features. Only their combination supports effective task learning.

