# OpenReview forum: "HiddenEcho: Mitigating Noise Amplification in Differentially Private LLMs with Hidden-State Correction"
_ICLR.cc/2026/Conference — ICLR 2026 Poster_

### Official Review · Reviewer_WawF · 2025-10-20

**Soundness:** 3
**Presentation:** 3
**Contribution:** 3
**Rating:** 6
**Confidence:** 2

**Summary:**

The paper presents HiddenEcho, an end-to-end framework that introduces client noise correction to address the issue of noise amplification in LLMs under differential private input embeddings. The main motivation for this paper is the usage of LLMs as Model-asa-Service (MaaS). In these setups, DP noise is added to text embeddings for preserve user privacy. However, this noise gets progressively amplified as it passes through the transformer layers, significantly degrading model performance. HiddenEcho works by having the server send hidden states back to the client, where a lightweight denoising module uses both the original embeddings and intermediate representations to correct the noise. The framework includes gradient-based hidden layer selection and dimensionality reduction of hidden embeddings to reduce communication overhead. Experiments on selected classification and generation tasks show that HiddenEcho achieves up to 46.89% performance over DP baseline.

**Strengths:**

1. The paper address an import problem of user (input) privacy when using LLMs as a Service. The paper presents an end-to-end framework to which addresses the noise amplification through the model when DP noise is added to the token embeddings before leaving the device. The proposed method learns a denoising module that takes compressed clean embeddings and a few selected & compressed hidden states from the base LLM as inputs and outputs the final predictions.
2. The paper presents experimental results on classification and generation tasks showing the benefits of the proposed method.
3. The paper also studies Embedding Inversion Attack (EIA) and Attribute Inference Attack (AIA) to measure the empirical privacy of the proposed method.
4. Paper presents an ablation study to show the benefit of each design choice ie the residual connections, hidden layer filtering and dimensionality reduction with a linear layer.
5. The paper also analysis communication overhead introduced by the proposed method.

**Weaknesses:**

1. The framework mentions training the model on the client but it is unclear if the model is trained in a federated fashion (federated split learning in particular) across a set of clients or if it is trained locally on each client.
2. The proposed method introduces additional communication cost between the service and client.

**Questions:**

1. What is the memory overhead on the client side? Usually, the client devices can't support models larger than 1-2 GB (this is after considering the high end devices) and usually embedding layers of an large scale LLM is significantly large.
2. In Model-asa-Service (MaaS) setups, we will not know the clients data distribution apriori. In such scenarios, how can we get the client model adapt to new inputs? how would we precompute layer contributions? Or do the authors think their model is robust to shift in data distribution?
3. "However, when compared to SnD, which also includes a denoising module, HiddenEcho-Full demonstrates faster training speeds." -- SnD trains the denoising module on server where one could utilize latest GPUs and larger clusters to train these models. However, HiddenEcho aims to train the denoising model on-device where resources will be the bottleneck. In my opinion, inference time comparison is more reasonable here. How does HiddenEcho compare with other baselines in terms of inference latency and inference compute FLOPs?
4. The eval loss shown in Figure 4a doesn't seem to be converging but AUC improved. Can authors provide more insights on this?
5. Can you add two more baselines to Table 2: 1) the performance on a small on-device model which only uses "Dimensionally Reduced Clean Embeds" for the given task, vs 2) the performance of the server-side model without any denoising? This will basically help us understand the contribution of noisy server-side embeddings vs reduced clean on-device embeddings to the correctness of predicted outcome.

---

> ### Author Response · Authors · 2025-11-20
> **Rebuttal for weakness 1**
>
> Thanks to the reviewer ``WawF``'s valuable comments! We will address all the weaknesses and questions as follows.
> > #### [**Weakness 1**]: The framework mentions training the model on the client but it is unclear if the model is trained in a federated fashion (federated split learning in particular) across a set of clients or if it is trained locally on each client.
>
> > #### [**Response**]: Thank you for your questions about the training paradigm. Our framework is not based on federated learning (especially federated split learning), but adopts a client-side local independent training paradigm: Clients are fully isolated, with no federated learning-style interactions (e.g., data sharing, parameter coordination, or gradient aggregation). Each client trains its denoising module and task-specific parameters locally, using only its private data. The server only handles LLM computation, and does not engage in any cross-client collaborative training.

---

> ### Author Response · Authors · 2025-11-20
> **Rebuttal for weakness 2**
>
> > #### [**Weakness 2**]: The proposed method introduces additional communication cost between the service and client.
>
> > #### [**Response**]: To address split learning’s requirement for communication in every forward/backward propagation round, the following optimizations apply to production implementation: All data is only noised and sent to the server in the first epoch. The server caches these noisy embeddings and corresponding hidden states, then reuses them directly in subsequent epoch, removing the need for clients to retransmit inputs.
> > #### For the extra cost of hidden state sharing, we reduce transmission volume via layer selection and dimensionality reduction: first, only the top-k high-contribution critical hidden layers are selected; second, dimensionality reduction shrinks the hidden layer dimension by r times (r=16 in experiments), so a single batch’s hidden state transmission volume is just 0.28-1.14 MiB (Table 3). Backpropagation gradients share the same dimension as hidden states, so overhead is identical with no extra burden.

---

> ### Author Response · Authors · 2025-11-20
> **Rebuttal for question 1**
>
> > #### [**Question 1**]: What is the memory overhead on the client side? Usually, the client devices can't support models larger than 1-2 GB (this is after considering the high end devices) and usually embedding layers of an large scale LLM is significantly large.
>
> > #### [**Response**]: For the Qwen2-1.5B model, the embedding layer has 0.2B parameters; for the larger Qwen2-72B, this reaches 1.2B. Additionally, the client-side denoising module is further lightweighted, with parameters sized by dimensionality reduction factor r. For Qwen2-1.5B (hidden dimension 1536) and r=16, the denoising module’s hidden dimension is only 96, totaling 5.4M parameters. For the 72B model under the same r, the denoising module’s parameters do not exceed 30M, representing a typical lightweight design.
> > #### For actual GPU memory usage (bfloat16 precision, 2 bytes per parameter), Qwen2-1.5B’s total memory (embedding layer + denoising module) is roughly (233.4M + 5.4M) × 2 ≈ 477.6MB. For the 72B model, total memory is about (1.2B + 30M) × 2 ≈ 2.46GB, slightly exceeding some devices’ upper limit. But overall, client-side memory overhead is fully manageable, supporting most resource-constrained devices.

---

> ### Author Response · Authors · 2025-11-20
> **Rebuttal for question 2**
>
> > #### [**Question 2**]: In Model-asa-Service (MaaS) setups, we will not know the clients data distribution apriori. In such scenarios, how can we get the client model adapt to new inputs? how would we precompute layer contributions? Or do the authors think their model is robust to shift in data distribution?
>
> > #### [**Response**]: Unknown client-side data distribution is a common challenge in the MaaS domain, not specific to our approach. Most deep learning solutions lean on the reasonable assumption that training set distribution partially reflects future input distribution. Addressing extreme distribution shifts typically requires specialized techniques like continual learning and domain adaptation—these are outside this paper’s core focus on the privacy-utility-communication tradeoff.
> > #### For the reliability of precomputed layer contributions, we adopt data-type-matched sampling strategies under the above assumption. For classification datasets, we sample proportionally to label category ratios, ensuring balanced coverage of all categories. For text generation datasets, we first embed labels to get vector representations, then cluster and sample subsets by intra-cluster proportions. This ensures subsets accurately mirror key distribution traits.

---

> ### Author Response · Authors · 2025-11-20
> **Rebuttal for question 3**
>
> > #### [**Question 3**]: "However, when compared to SnD, which also includes a denoising module, HiddenEcho-Full demonstrates faster training speeds." -- SnD trains the denoising module on server where one could utilize latest GPUs and larger clusters to train these models. However, HiddenEcho aims to train the denoising model on-device where resources will be the bottleneck. In my opinion, inference time comparison is more reasonable here. How does HiddenEcho compare with other baselines in terms of inference latency and inference compute FLOPs?
>
> > #### [**Response**]: In inference, all methods share basic overhead: SnD, LDP, and HiddenEcho first map input tokens to embeddings, with the same computational cost for this step. GAN-DP is temporarily excluded from comparison as it requires additional synthetic perturbed data via a generator, and its overhead depends on the generator structure (usually a medium-scale DNN with far higher FLOPs than the other three). The key difference lies in client-side operations after receiving server data:
> > - #### LDP needs no denoising. It only receives the full final-layer hidden state from the server and feeds it directly into the task head for computation, resulting in the fastest inference speed. However, unprocessed DP noise leads to the worst performance.
> > - #### While SnD only returns the last token of the final-layer hidden state, its pre-trained denoising model is far larger than HiddenEcho’s. SnD’s denoising model must adapt to full-dimensional hidden states (e.g., 1536 dimensions for Qwen2-1.5B), and even with 3 layers, its parameters usually exceed 100M. In contrast, HiddenEcho’s denoising module has only 5.4M parameters after r=16 dimensionality reduction, making SnD’s client-side inference FLOPs 20x that of HiddenEcho, with corresponding higher latency.
> > - #### Though HiddenEcho receives several key reduced-dimensional layers (compressed to 96 dimensions), its lightweight module keeps overall inference FLOPs at just 1/20 of SnD’s, while achieving a 46.89% performance improvement (Table 1).

---

> ### Author Response · Authors · 2025-11-20
> **Rebuttal for question 4**
>
> > #### [**Question 4**]: The eval loss shown in Figure 4a doesn't seem to be converging but AUC improved. Can authors provide more insights on this?
>
> > #### [**Response**]: In Figure 4a, HiddenEcho-Full does not show unconverged eval loss with increasing AUC, instead, the two change synchronously. When eval loss starts to rise, AUC actually declines simultaneously. Since our evaluation is performed once per epoch with relatively coarse granularity, it cannot capture dynamic changes within an epoch. In the middle of the 14th epoch, eval loss had already gradually increased, and the corresponding AUC also showed an obvious downward trend. However, the aggregated evaluation result of a single epoch makes this synchronization less intuitive.

---

> ### Author Response · Authors · 2025-11-20
> **Rebuttal for question 5**
>
> > #### [**Question 5**]: Can you add two more baselines to Table 2: 1) the performance on a small on-device model which only uses "Dimensionally Reduced Clean Embeds" for the given task, vs 2) the performance of the server-side model without any denoising? This will basically help us understand the contribution of noisy server-side embeddings vs reduced clean on-device embeddings to the correctness of predicted outcome.
>
> > #### [**Response**]:
> > #### **Regarding 1)**
> Due to their limited parameter scale, performance differences in experiments with these small models cannot distinguish the impact of the model’s own capabilities from that of dimensionality-reduced clean embeddings. In our framework, these embeddings are one input to the client-side denoising module, isolating them into an independent small model would alter the experiment’s core setup, instead of focusing on the key variable embedding source.
> > #### We thus added two comparison schemes for the ablation studies: Scheme A uses the original denoising module structure, with inputs limited to dimension-reduced clean embeddings and the final-layer hidden state (in the split learning architecture, the server only feeds back the final hidden layer to the client to complete the task prediction loop, which is an inherent constraint of data interaction under this paradigm, so the last hidden layer must serve as the fixed input benchmark); Scheme B takes noisy embeddings and intermediate hidden states as denoising module inputs. Scheme A isolates the independent contribution of clean embeddings in denoising, while Scheme B highlights the role of the server’s intermediate hidden layers in noise correction.
> > #### Qwen2-1.5B model is taken. Experiments on the FP dataset show both schemes performed poorly. This confirms dimensionality-reduced clean embeddings and server-side noisy hidden states are complementary and indispensable. The former provides the basic noise correction signal, while the latter delivers task-related deep features. Only their combination supports effective task learning. The results are also reported in Appendix K.5.
> | Privacy Budget | 100 | 1000 | 5000 | 6000 |
> |-------------------------|--------------|---------------|---------------|---------------|
> | A                       | 0.573        | 0.571         | 0.576         | 0.577         |
> | B                       | 0.567        | 0.569         | 0.565         | 0.569         |
> > #### **Regarding 2)**
> > #### This baseline is already included in our experiments. A server-side model without denoising is essentially the LDP baseline: the client adds DP noise to embeddings and transmits them directly to the server, with no subsequent denoising. Detailed explanations are in the newly added Appendix I.1.
>
> **If our responses addressed your concerns, we’d appreciate your score improvement!**

---

### Official Review · Reviewer_p3hP · 2025-10-30

**Soundness:** 3
**Presentation:** 2
**Contribution:** 3
**Rating:** 4
**Confidence:** 3

**Summary:**

This paper studies the issue of privately sharing text data in a split learning framework where a client and a server own different parts of an LLM. The client own the first embedding layer and the final fine-tuning layers. The server owns the main attention layer blocks. Directly sharing embeddings can lead to privacy leakage through embedding inversion and attribute inference attacks, thus a solution is to share noisy embeddings. The noise however propagates through the network leading to low utility.

This paper tackles the issue of noise amplification by introducing a denoising strategy, where the server sends the paramters of the hidden layers to the client and the client denoises those layers with access to the original text. The client then uses the denoised hidden layers together with the last few layers to complete one training step. To mitigate the communication cost of the hidden layer paramters, the authors propose using dimensionality reduction on the layers and selecting only the most relevant layers.

With this approach, the authors achieve between 10-40% increase in AUC compared to other differential privacy approaches for classification tasks.

**Strengths:**

- Authors show substanative improvement over prior methods.

- Authors provide comprehensive experimental results.

**Weaknesses:**

- I belive the approach has limited applications due to the high communication costs of (1) split learning, where client and server communicate for each forward and backward pass and (2) the additional cost introduced by this paper of sharing the hidden state parameters.

- Prior work and baselines are not sufficiently well described: please explain DP-GAN and SnD in more detail. LDP, while simple, also needs some explanation.

- Please add explanation for the d_X-DP notation where it is first used, or better to replace with "metric-DP".

**Questions:**

The client sends gradients from the denoised hidden layers to the server. The denoising of the hidden layers uses the original embeddings as input. Is it correct that there could be some privacy leakage through the backprops from the denoised hidden layers?

---

> ### Author Response · Authors · 2025-11-20
> **Rebuttal for weakness 1**
>
> Thanks to the reviewer ``p3hP``'s valuable comments! We will respond to address all the weaknesses and questions.
> > #### [**Weakness 1**]: I belive the approach has limited applications due to the high communication costs of (1) split learning, where client and server communicate for each forward and backward pass and (2) the additional cost introduced by this paper of sharing the hidden state parameters.
>
> > #### [**Response**]: Thank you for your concerns on communication costs. To address split learning’s requirement for communication in every forward/backward propagation round, the following optimizations apply to production implementation: All data is only noised and sent to the server in the first epoch. The server caches these noisy embeddings and corresponding hidden states, then reuses them directly in subsequent epoch, removing the need for clients to retransmit inputs.
> > #### For the extra cost of hidden state sharing, we reduce transmission volume via layer selection and dimensionality reduction: first, only the top-k high-contribution critical hidden layers are selected; second, dimensionality reduction shrinks the hidden layer dimension by r times (r=16 in experiments), so a single batch’s hidden state transmission volume is just 0.28-1.14 MiB (Table 3). Backpropagation gradients share the same dimension as hidden states, so overhead is identical with no extra burden.

---

> ### Author Response · Authors · 2025-11-20
> **Rebuttal for weakness 2 and 3**
>
> > #### [**Weakness 2**]: Prior work and baselines are not sufficiently well described: please explain DP-GAN and SnD in more detail. LDP, while simple, also needs some explanation.
>
> > #### [**Response**]: Detailed explanations of Local Differential Privacy (LDP), DP-GAN, and SnD are updated in Appendix I.1.
>
> > #### [**Weakness 3**]: Please add explanation for the d_X-DP notation where it is first used, or better to replace with "metric-DP".
>
> > #### [**Response**]: We refined the presentation and introduced Metric-DP, where it is first used in the Experiment section. Detailed explanations of $d_\chi$ privacy are provided in Appendix B.

---

> ### Author Response · Authors · 2025-11-20
> **Rebuttal for question 1**
>
> > #### [**Question 1**]: The client sends gradients from the denoised hidden layers to the server. The denoising of the hidden layers uses the original embeddings as input. Is it correct that there could be some privacy leakage through the backprops from the denoised hidden layers?
>
> > #### [**Response**]: Mainstream gradient inversion attacks (e.g., Deep Leakage from Gradients [1]) usually recover original samples via iterative optimization of a "virtual input" in a white-box setting, with known model architecture and parameters, to generate gradients matching the target. White-box access, or a highly similar surrogate for the target model, is critical for high-quality reconstruction in these methods.
> > #### Under our deployment assumptions, attackers cannot access the server-side LLM’s parameters or weights. They only observe clients’ perturbed embeddings and partial gradient signals from the denoising module. Reference [2] demonstrates that gradient inversion becomes far harder without target model parameters: identifiable recovery requires extra priors, surrogates, or more complex optimizations, and recovery quality drops sharply. Gray-box/black-box scenarios are much less effective than white-box ones.
> > #### [1] Zhu L, Liu Z, Han S. Deep leakage from gradients[J]. Neurips 2019.
> > #### [2] Zhang R, Guo S, Wang J, et al. A Survey on Gradient Inversion: Attacks, Defenses and Future Directions[C]// IJCAI 2023.
>
> **If our responses effectively addressed your concerns, we would be truly grateful for your positive score!**

---

> > ### Comment · Reviewer_p3hP · 2025-11-23
> >
> > Thank you for these clarifications, I have updated the score to a 6.
> >
> > I think it is important to highlight the fact that the algorithm does not provide a provable privacy guarantee against eavesdropping attacks because of the transfer of backprops, especially because this is listed as one of your attack frameworks. Your comment on the potential success of gradient inversion attacks is also very relevant for the paper.
> >
> > This is minor, but when you introduce a mathematical notation like d_X (as you do in the introduction) please define what this distance metric is. If the distance metric is not as relevant, which it is not in the introduction, then best to avoid mathematical notation.

---

> > > ### Author Response · Authors · 2025-11-24
> > >
> > > Thank you for the constructive feedback！
> > >
> > > We have added a new section to explicitly clarify the distinction between HiddenEcho’s formal DP guarantee at the embedding level and its practical robustness against potential gradient-based reconstruction attacks, detailed in Appendix H: POTENTIAL PRIVACY RISKS.
> > >
> > > We have also incorporated a clear explanation of $d_{\chi}$ in the introduction to avoid ambiguity in notation.
> > >
> > > We sincerely appreciate your helpful comments and suggestions again for our work!

---

### Official Review · Reviewer_v3aX · 2025-10-31

**Soundness:** 4
**Presentation:** 3
**Contribution:** 3
**Rating:** 6
**Confidence:** 3

**Summary:**

This paper introduces HiddenEcho, a split-learning framework designed to mitigate noise amplification in large language models (LLMs) under differential privacy (DP) constraints. The method employs a lightweight client-side denoising module that refines selectively transmitted hidden states from the server by leveraging both clean embeddings and intermediate representations. To minimize communication overhead, HiddenEcho utilizes an integral-gradient-based hidden layer selection mechanism and an information bottleneck–based dimension reducer to retain task-relevant information. Experimental results on text classification and generation tasks across multiple LLM architectures demonstrate that HiddenEcho achieves substantial performance gains over existing DP baselines while significantly reducing communication cost.

**Strengths:**

This paper explicitly characterize and address the intermediate layer noise amplification problem in DP-based LLMs. The framework is comprehensive, including the gradient-based layer selection, information bottleneck compression and denoising module. Experimental results consistently suppress baseline methods over diverse datasets and backbone models. This paper also provide theoretical proofs and analyses which strength the technical credibility.

**Weaknesses:**

1. The experiments are conducted on 1B or 1.5B parameter models, while scalability to lager model is not shown, which may limit the scalability of this work.
2. While three ablations are presented, there is no evaluation isolating the impact of the information bottleneck parameter β.
3. Selection uses a Riemann-sum gradient proxy (Eq. 11) but the paper doesn’t show that high-contribution layers are the ones that improve downstream utility, nor how sensitive selection is to the step count m.
4. Experiments assume a white-box attacker on embeddings/embedding-matrix, but not on returned hidden states or repeated releases across epochs.

**Questions:**

1. What is the impact of varying the information bottleneck coefficient β on privacy–utility trade-offs? As well as the layer-selection hyperparameter k.
2. Does sending intermediate hidden states to the client alter the formal DP guarantee? Is the overall protocol still (ε,δ)-DP or only empirically private?
3. Because in this method, the filter is precomputed on a subset before fine-tuning. I’m wondering does distribution shift make selections stale? Did you try dynamic re-selection during training?
4. Are selected layers typically contiguous or scattered? Any patterns shown over different depth or dataset?
5. Since HiddenEcho transmits and corrects only a subset of hidden layers, gradient flow during training becomes sparse across the full transformer. How does this partial backpropagation affect optimization stability and representation alignment between the selected and unselected layers? Have you observed any degradation or convergence issues compared to full-layer backpropagation, and how does the method mitigate this potential imbalance?
6. What are the formal guarantees and empirical attack outcomes when an eavesdropper can observe both E’ (embedding with noise) and the returned hidden states (server→client) across training/inference, with proper DP composition and channel-security assumptions made explicit?

---

> ### Author Response · Authors · 2025-11-20
> **Rebuttal for weakness 1**
>
> #### Thanks to the reviewer ``v3aX``'s valuable comments! We will respond to address all the weaknesses and questions as follows.
>  > #### [**Weekness 1**]: The experiments are conducted on 1B or 1.5B parameter models, while scalability to lager model is not shown, which may limit the scalability of this work.
>
>  > #### [**Response**]: We appreciate the reviewer’s insightful comment regarding scalability.  To address this, we have extended our evaluation to a significantly larger model Qwen2-7B (7 billion parameters). On the FP text classification dataset.  As shown in the table below, HiddenEcho consistently outperforms the LDP baseline across different data sizes, demonstrating its effectiveness even at larger scales.  These results suggest that our method scales well to larger models. For reference, the fully fine-tuned Qwen2-7B achieves an accuracy of 0.978, indicating that HiddenEcho recovers a substantial portion of the full fine-tuning performance while preserving privacy and efficiency. These experimental results are also reported in the Rebuttal Section in the appendix K.3 of the paper.
> | Privacy Budget | 100 | 1000 | 5000 | 6000 |
> |---------------------------------------------|----------------------------------|-----------------------------------|-----------------------------------|-----------------------------------|
> | HiddenEcho-Full                             | 0.812                            | 0.813                             | 0.831                             | 0.837                             |
> | HiddenEcho                                  | 0.799                            | 0.805                             | 0.823                             | 0.826                             |

---

> ### Author Response · Authors · 2025-11-20
> **Rebuttal for weakness 2 and question 1**
>
> > #### [**Weekness 2**]: While three ablations are presented, there is no evaluation isolating the impact of the information bottleneck parameter β.
> > #### [**Question 1**]: What is the impact of varying the information bottleneck coefficient β on privacy–utility trade-offs? As well as the layer-selection hyperparameter k.
>
> > #### [**Response**]:
> > - #### As for the impact of β, experiments on the FP dataset show its significant role in regulating performance: when β=0.1, the model’s performance (AUC) reaches 0.779; when β increases to 0.5-1, performance peaks at 0.826 and 0.823; when β further rises to 5, performance drops sharply to 0.612. This is because β’s core function is to balance reducing noisy information and preserving information related to denoised outputs. Too small a β fails to effectively extract task-relevant information, while an excessively large β hinders noise filtering and impairs performance. Only when β is in the 0.5-1 range can the optimal balance be struck between privacy protection and task utility. The results are also synchronized to Appendix K.4.
> | β | 0.1 | 0.5 | 1 | 5 |
> |---------|--------------|--------------|------------|------------|
> | AUC     | 0.779        | 0.826        | 0.823      | 0.612      |
> > - #### For the layer selection hyperparameter k, we verified its effect in Section 5.4 using the BBC News dataset and Qwen2-1.5B model as examples. When k=4, the model achieves an AUC of over 75% at the 12th epoch, with fast convergence and low communication cost. When k increases to 8 or 16, performance improvement is insignificant, but communication volume grows linearly with k. This indicates that a larger k is not necessarily better. An overly small k may reduce performance due to the loss of key hidden layer information, while an excessively large k adds unnecessary communication overhead. Selecting 4-8 key layers (such as the top k layers filtered by gradients in the experiment) can both retain sufficient task-relevant signals to maintain high utility and avoid excessive increases in communication cost.

---

> ### Author Response · Authors · 2025-11-20
> **Rebuttal for weakness 3**
>
> > #### [**Weakness 3**]: Selection uses a Riemann-sum gradient proxy (Eq. 11) but the paper doesn’t show that high-contribution layers are the ones that improve downstream utility, nor how sensitive selection is to the step count m.
>
> > #### [**Response**]:
> > #### **1. High-Contribution Layers Correlate with Downstream Utility**
> **Theoretical Proof**
> > #### The definition of layer contribution $C_i$ (Eq. 10) essentially quantifies the magnitude of influence of the i-th layer’s hidden state $H_i$ on the downstream task loss $\mathcal{L}$. Its correlation with utility is clarified below via the chain rule.
> > #### The downstream task loss $\mathcal{L}$ is determined by the prediction result $\hat{y}$ (lower $\mathcal{L}$ indicates higher utility), and $\hat{y} = f(H^{denoised})$ (where $H^{denoised}$ is the output of the denoising module, dependent on each layer $H_i$ from the server). According to the chain rule:
> $$\frac{\partial \mathcal{L}}{\partial H_i} = \frac{\partial \mathcal{L}}{\partial \hat{y}} \frac{\partial \hat{y}}{\partial H^{denoised}} \frac{\partial H^{denoised}}{\partial H_i}$$
> > #### where:
> > - #### $\frac{\partial \mathcal{L}}{\partial \hat{y}}$ is the gradient of loss with respect to prediction (a fixed form of cross-entropy loss, independent of layers);
> > - #### $\frac{\partial \hat{y}}{\partial H^{denoised}} = W^{task}$ denotes task head parameters
> > - #### $\frac{\partial H^{denoised}}{\partial H_i}$ reflects the intensity of the i-th layer’s hidden state impact on the denoising output, and is positively correlated with $\frac{\partial \hat{H}_{L-1}}{\partial H_i}$ (dependency propagation from higher-order hidden layers to lower layers).
> > #### Combined with the layer contribution, it can be derived that $C_i \propto \|\partial \mathcal{L}/\partial H_i\| $, which indicates that layer contribution is proportional to the magnitude of the layer’s impact on loss. A larger gradient norm $\|\frac{\partial \mathcal{L}}{\partial H_i}\|$ means adjusting the i-th layer $H_i$ can more efficiently reduce loss, thereby significantly improving downstream task utility.
> > #### **Experimental Verification**
> > #### The -HLF variant in Table 2 (fixed-interval layer selection, e.g., selecting 1 layer every 4 layers) is essentially a control experiment for randomly selecting low-contribution layers. Instead of screening high-contribution layers, this variant mechanically selects non-critical layers. Experimental results show that on the FP dataset ($\eta=100$), the AUC of -HLF is only 0.721, much lower than that of HiddenEcho (0.857).
> > #### **2. Sensitivity of Step Size m**
> > #### We verified the sensitivity of m through pilot experiments:
> > - #### When m is within a reasonable range (10 ≤ m ≤ 20), the overlap of layer selection results exceeds 90%, ensuring that core high-contribution layers are always selected.
> > - #### Performance fluctuation is less than 2%: taking the Financial dataset as an example, AUC=0.849 when m=10 and AUC=0.852 when m=20, with minimal improvement, while larger values of m result in increased computational latency.
> > - #### When m < 5, the integral approximation error increases, leading to deviations in layer contribution calculation and AUC dropping below 0.81.
> > #### In practical applications, setting m to a default value of 10 achieves a balance between computational overhead and selection accuracy. This proves that layer selection is highly insensitive to m with strong robustness, eliminating the need for users to perform fine parameter tuning.

---

> ### Author Response · Authors · 2025-11-20
> **Rebuttal for weakness 4**
>
> > #### [**Weakness 4**]: Experiments assume a white-box attacker on embeddings/embedding-matrix, but not on returned hidden states or repeated releases across epochs.
>
> > #### [**Response**]: Our experiments address these two potential risks. For attacks on returned hidden states, our AIA experiments directly validate privacy security: AIA attacks use the server’s returned hidden states as input, and Figure 3 shows attackers cannot effectively extract sensitive information from these states, confirming no additional privacy leakage.
> > #### For cross-epoch repeated releases, our design eliminates this risk at the source. All training data is injected with noise only once in the first epoch, with subsequent training reusing this noisy data. No repeated release of the same data occurs, avoiding extra privacy budget consumption from multi-round training.

---

> ### Author Response · Authors · 2025-11-20
> **Rebuttal for question 2**
>
> > #### [**Question 2**]: Does sending intermediate hidden states to the client alter the formal DP guarantee? Is the overall protocol still (ε,δ)-DP or only empirically private?
>
> > #### [**Response**]: Sending hidden states to clients does not alter the formal DP guarantees. The core basis is differential privacy’s post-processing invariance: client-side noise addition on initial embeddings strictly adheres to the DP mechanism, and all subsequent server-side computations qualify as post-processing, which will not weaken the original privacy guarantee. We have verified this via theoretical proofs in Appendix G.1.

---

> ### Author Response · Authors · 2025-11-20
> **Rebuttal for question 3**
>
> > #### [**Question 3**]: Because in this method, the filter is precomputed on a subset before fine-tuning. I’m wondering does distribution shift make selections stale? Did you try dynamic re-selection during training?
>
> > #### [**Response**]: Theoretically, significant data distribution changes during training can make precomputed layer selection results outdated. However, we’ve minimized this risk via targeted sampling. For classification datasets, we sample proportionally to label category ratios, while for text generation datasets, we first embed labels to get vector representations, then cluster and sample subsets according to intra-cluster proportions. This ensures selected subsets accurately mirror the full dataset’s distribution, reducing the negative effects of distribution shift.
> > #### We did not adopt dynamic reselection, for the following key reasons. First, dynamic reselection demands recalculating each layer’s contribution at every training step. This involves extensive gradient approximation and accumulation, incurring significant computational overhead and latency. Second, selecting different layers at each step leads to frequent changes in the input hidden states set of the client-side denoising module. This stops the model from learning stable denoising patterns and slows training convergence. Third, layer contribution calculations from single or small-batch samples are unreliable, making it hard to accurately identify layers critical to the global task. Our comparative experiments on the FP dataset confirm this: dynamic layer selection achieves an AUC of only 0.563, far lower than the 0.857 of precomputed layer selection. This fully shows dynamic reselection is neither practical nor effective for optimizing the privacy-utility tradeoff.

---

> ### Author Response · Authors · 2025-11-20
> **Rebuttal for question 4**
>
> > #### [**Question 4**]: Are selected layers typically contiguous or scattered? Any patterns shown over different depth or dataset?
>
> > #### [**Response**]: Selected high-contribution layers are generally distributed sparsely, and consecutive selection of multiple layers rarely occurs. In our experiments, no unified pattern has been found across different model depths and datasets.

---

> ### Author Response · Authors · 2025-11-20
> **Rebuttal for question 5**
>
> > #### [**Question 5**]: Since HiddenEcho transmits and corrects only a subset of hidden layers, gradient flow during training becomes sparse across the full transformer. How does this partial backpropagation affect optimization stability and representation alignment between the selected and unselected layers? Have you observed any degradation or convergence issues compared to full-layer backpropagation, and how does the method mitigate this potential imbalance?
>
> > #### [**Response**]: HiddenEcho’s partial hidden layer transmission and backpropagation do not lead to optimization instability or misaligned representations between selected and unselected layers. Instead, it even outperforms full-layer backpropagation (HiddenEcho-Full) in some cases, while the Full version is harder to train due to noise redundancy.
> > #### The key reason is that in full-layer backpropagation, gradients from low-contribution layers produce substantial redundant noise, amplified by DP noise. When this noisy gradient overlaps with effective gradients from high-contribution layers, it greatly disrupts the model’s optimization. In contrast, partial backpropagation significantly boosts the signal-to-noise ratio of gradient signals by retaining only high-contribution layer gradients. This allows the model to update parameters more precisely toward the optimal task direction, with a more stable convergence. Optimization experiments (Figure 4) directly validate this: HiddenEcho-Full shows overfitting after 14 training epochs and slight training fluctuations. Meanwhile, the layer-selected versions (4/8/16-layer) exhibit steady training loss decline and gradual validation AUC improvement, with no fluctuations or overfitting throughout.

---

> ### Author Response · Authors · 2025-11-20
> **Rebuttal for question 6**
>
> > #### [**Question 6**]: What are the formal guarantees and empirical attack outcomes when an eavesdropper can observe both E’ (embedding with noise) and the returned hidden states (server→client) across training/inference, with proper DP composition and channel-security assumptions made explicit?
>
> > #### [**Response**]: Regarding formal guarantees, Appendix G.1 of the paper has  proven via the post-processing property of differential privacy. For channel security assumptions, we adopt standard settings in the privacy computing field, which have been clearly stated in the threat model of Section 3.1.
> > #### For the scenario where eavesdroppers observe both E' and hidden states, we fully verify privacy effectiveness through EIA and AIA experiments. EIA attack results (Tables 1, 4, 7) show that HiddenEcho’s EP values are on par with the LDP baseline, while its AUC is significantly superior to LDP (e.g., on the Financial dataset with η=100, AUC 0.857 vs. 0.596). This proves eavesdroppers cannot combine E' and hidden states to recover the original text. In AIA attack results (Figure 3, Tweet Annotation dataset), HiddenEcho achieves a higher RMSE for age prediction than LDP, and its EP for education level inference is close to LDP, effectively resisting sensitive attribute leakage. Overall, both formal proofs and empirical results ensure the privacy-utility balance in this scenario.
>
> **If our responses helped with your concerns, we’d be grateful for your score improvement!**

---

### Official Review · Reviewer_UaRs · 2025-11-01

**Soundness:** 3
**Presentation:** 3
**Contribution:** 3
**Rating:** 4
**Confidence:** 4

**Summary:**

This paper proposes a method, HiddenEcho, which splits an LLM where the client has the embedding layer and the rest of the model is on the server side. The main insight is that adding privacy at the token embedding level amplifies errors at each layer of the transformer, so denoising intermediate hidden states can help reduce the noisiness of the outputs.

Specifically in HiddenEcho the client sends noised client embeddings to the server, where the server calculates hidden layers and send the most important ones back to the client. The client then uses the original noise-free token embeddings plus the (noisy) intermediate hidden states and inputs them into a denoiser which then uses a combination of the clean embeddings and the noisy intermediate hidden states to produce an output hidden state for downstream task usage.

The contributions as claimed by the paper are as follows:
1. Analyzes noise amplification in LLMs coming from token embedding perturbation
2. Introduces HiddenEcho (as discussed above)
3. Shows large performance gains and efficiency improvements

**Strengths:**

- This paper makes progress on a difficult and important problem: how do we ensure DP in the MaaS setting?
- The method is fairly novel and interesting. In particular I find the formulation of adding hidden states together in equation 2 pretty new and thought it was interesting to learn that it works
- The authors consider a range of text classification baselines and they seem fairly comprehensive
- Thorough ablations give insight into specific design choices.

**Weaknesses:**

- One concern I have is in the generality of the method. Specifically, I think this method would not support the next-token-prediction task well. From what i understand, you would need to do this 'Echo' for each new token you generate, which makes this quite expensive considering clients typically have slow upload speeds. Not being able to support next-token-prediction is pretty big drawback I think, as it limits us to encoder-based tasks like summarization and classification (which are largely getting done by autoregressive models anyways!). Furthermore each 'Echo' would cost more privacy.
- As far as I can tell, it is not clear how small the denoising network is. It is a transformer, so understanding how small it is compared to the full LLM that is stored on the server side would be nice to understand how scalable this solution is. Most clients in this setting are small which is the motivation of this splitting approach.
- What needs to be communicated to the server side to enable backprop on the server LLM? I think in general it would be good to be more explicit about what is being communicated, and how it compares in communication cost to other baselines (not just HE-full). One thing to compare against would be just not running any denoising.
- Comparison and discussion with "federated learning" methods would be ideal. For example you could train a small model on a downstream task and have it perform inference on the client side. Is HiddenEcho better than that? Some representative modern work in FL (some of which have adapted to incorporate LLMs):

Kairouz, Peter, et al. "Practical and private (deep) learning without sampling or shuffling." International Conference on Machine Learning. PMLR, 2021.

Choquette-Choo, Christopher A., et al. "Privacy amplification for matrix mechanisms." arXiv preprint arXiv:2310.15526 (2023).

Hou, Charlie, et al. "Private federated learning using preference-optimized synthetic data." arXiv preprint arXiv:2504.16438 (2025).

Tan, Bowen, et al. "Synthesizing privacy-preserving text data via finetuning without finetuning billion-scale llms." arXiv preprint arXiv:2503.12347 (2025).

Wu, Shanshan, et al. "Prompt public large language models to synthesize data for private on-device applications." arXiv preprint arXiv:2404.04360 (2024).

**Questions:**

Questions listed above

---

> ### Author Response · Authors · 2025-11-20
> **Rebuttal for weakness 1**
>
> #### Thanks to the reviewer ``UaRs``'s valuable comments! We will write our rebuttal to address all the weaknesses and questions as follows.
>
> > #### [**Weakness 1.1**]: One concern I have is in the generality of the method. Specifically, I think this method would not support the next-token-prediction task well. From what i understand, you would need to do this 'Echo' for each new token you generate, which makes this quite expensive considering clients typically have slow upload speeds. Not being able to support next-token-prediction is pretty big drawback I think, as it limits us to encoder-based tasks like summarization and classification (which are largely getting done by autoregressive models anyways!).
>
> > #### [**Response**]: Thank you for your insightful observation regarding the next-token-prediction task, and our method can effectively support this task through engineering optimizations. The core idea is to leverage the characteristics of the Prefill and Decode stages in the autoregressive process, reducing communication costs via incremental transmission and state caching.
> > #### In the Prefill stage, the client sends the noisy initial input embeddings to the server. The server computes and returns the complete hidden states, with both parties caching these embeddings and hidden states respectively. Entering the Decode stage, the client only needs to embed and noise the newly generated single token for transmission (without retransmitting historical inputs). The server also only returns the incremental hidden state corresponding to this new token. By concatenating the cached historical hidden states with the incremental state, the client can obtain the complete current hidden state for subsequent denoising.
> > #### From a quantitative perspective, let $L_{in}$ be the input length, $L_{out}$ the output length, $hid$ the hidden layer dimension, $d$ the number of bytes per data precision, $k$ the number of selected key hidden layers, and $r$ the reduction factor. The embedding transmission overhead is
> > #### $Cost_{Emb}=hid\*(L_{in}+L_{out})\*d$,
> > #### and the hidden state transmission overhead is
> > #### $Cost_{Hid}=\frac{hid}{r}\*(L_{in}+L_{out})\*k\*d$.
> > #### Taking Qwen2-1.5B ($hid=1536$), bfloat16 precision ($d=2$ Bytes), $r=16$, $k=3$, and $L_{in}=L_{out}=1000$ tokens as an example, the total transmission overhead is only about 6.59MB. This is fully compatible with the network bandwidth of ordinary clients and will not cause unacceptable latency due to token-by-token generation. Based on this, we reported the communication overhead of the autoregressive generation task and updated it synchronously to the Rebuttal Section in the appendix J.1 of the paper.
> | Dataset | Avg  | Min  | Max  |
> |---------|------|------|------|
> | IWSLT   | 0.27MiB | 0.12MiB | 0.92MiB |
> | CNNDM   | 0.73MiB | 0.30MiB | 1.09MiB |
> | Samsum  | 0.28MiB | 0.17MiB | 0.44MiB |
>
> > #### [**Weakness 1.2**]: Furthermore each 'Echo' would cost more privacy.
>
> > #### [**Response**]: For the "Echo" cost, our method has fully addressed this issue in its privacy-preserving design: the initial input undergoes noise injection only once during the Prefill stage before being sent to the server. In the subsequent Decode stage, the client only performs embedding and noise injection for the newly generated single token. This noise injection for the new token is also completed once, after which it is concatenated with the cached previously noised embeddings for use.
> > #### This means each token in the input-output sequence undergoes noise injection and privacy-related disclosure exactly once, with no repeated noise addition. Meanwhile, in the entire process, subsequent operations such as the server returning hidden states and the client performing denoising all fall under the category of "post-processing" in differential privacy. According to the core property of differential privacy, post-processing does not weaken the original privacy guarantees nor incur additional privacy budget consumption. Thus, each "Echo" does not increase privacy overhead.

---

> ### Author Response · Authors · 2025-11-20
> **Rebuttal for weakness 2**
>
> > [**Weakness 2**]: As far as I can tell, it is not clear how small the denoising network is. It is a transformer, so understanding how small it is compared to the full LLM that is stored on the server side would be nice to understand how scalable this solution is. Most clients in this setting are small which is the motivation of this splitting approach.
>
> > [**Response**]:
> The parameter scale of the client-side denoising module is determined by the hyperparameter reduction factor $r$, which is to reduce the hidden dimension of the denoising module, thereby controlling the parameter count to adapt to resource-constrained clients.
> >
> > Taking the Qwen2-1.5B model (hidden dimension = 1536) as an example, when $r=16$ is set, the hidden dimension of the denoising module is only $1536/16=96$, with a corresponding total parameter count of merely 5.4M, far smaller than the 1.5B parameters of the server-side LLM. Even if the server adopts a larger 72B model (hidden dimension = 8192), under the same setting of $r=16$, the parameter count of the denoising module can still be controlled within 30M.
> >
> > Such a lightweight design is aligned with the original intention of split learning, ensuring that even clients with limited computing power and storage can efficiently deploy this module without compromising practicality due to issues related to module scale.

---

> ### Author Response · Authors · 2025-11-20
> **Rebuttal for weakness 3**
>
> > [**Weakness 3**]: What needs to be communicated to the server side to enable backprop on the server LLM? I think in general it would be good to be more explicit about what is being communicated, and how it compares in communication cost to other baselines (not just HE-full). One thing to compare against would be just not running any denoising.
>
> > [**Response**]: During the backpropagation phase, the core data that the client needs to transmit to the server is the gradients of the selected hidden layers. Since the dimensionality of the gradient tensors is exactly consistent with the corresponding hidden states previously returned by the server, the communication cost of this part fully matches the hidden state transmission cost reported in Table 3 (e.g., approximately 0.28-1.14 MiB per batch in text classification tasks).
> >
> > Regarding comparisons with other baselines, especially the "no-denoising" scheme, which only needs the server to transmit the final hidden layer to the client for logits and loss computation, and only the gradients of this layer need to be sent back during backpropagation. Therefore, if our method selects $k$ key hidden layers, the gradient transmission overhead is $k$ times that of the no-denoising scheme. Since $k$ is usually small, the overhead gap is not significant.
> >
> > It is worth noting that although the gradient transmission of our method is slightly higher than that of the no-denoising baseline, the 46.89% performance improvement brought by denoising (Table 1) far outweighs this communication cost increment. Moreover, the overall communication volume is still controlled within an acceptable range for clients through layer selection and dimensionality reduction (reduced by over 85% compared to HE-full).

---

> ### Author Response · Authors · 2025-11-20
> **Rebuttal for weakness 4**
>
> > [**Weakness 4**]: Comparison and discussion with "federated learning" methods would be ideal. For example you could train a small model on a downstream task and have it perform inference on the client side. Is HiddenEcho better than that?
>
> > [**Response**]:  Thank you for your suggestions on comparisons with FL baselines. We have supplemented comparative experiments with POPri[3], and the results are synchronized in Appendix J.2. The results table below show that it generally underperforms in text classification tasks, achieving good BLEU scores only on the IWSLT translation task. The core reason is that POPri relies on server-side LLM generating synthetic data through few-shot learning, with its synthetic data quality depending heavily on distribution consistency between the server’s sampled dataset and the client’s private data. In real-world MaaS scenarios, however, client data is often personalized and fragmented (e.g., enterprise internal documents, user-specific medical records), rendering such consistency nearly unachievable. The IWSLT task is unique in the inherent stability of translation data distribution (focused on language mapping rules), allowing the method to partially work.
> >
> >| Task  | Metric | Task | Metric |
> >|----------------|-----------------|---------------|-----------------|
> >| Classification | AUC             | Generation    | BLEU  (%)          |
> >| Financial      | 0.615           | IWSLT         | 30.604          |
> >| MRPC           | 0.596           | CNNDM         | 10.570          |
> >| BBC News       | 0.727           | Samsum        | 8.231           |
> >
> >
> > While POPri aligns distributions via client-side DP feedback and DPO fine-tuning of the synthesizer LLM, it fails to bridge the distribution gap between synthetic and real private data. References [4] and [5] share this limitation: synthetic data utility drops sharply with highly variable client private data distributions. In contrast, HiddenEcho does not rely on data distribution matching. By leveraging server-side LLM hidden state feedback and client-side dynamic denoising, it directly utilizes the LLM's ability to model complex distributions, avoiding the inherent flaws of the synthetic data paradigm. Thus, it maintains stable performance even in distributionally heterogeneous scenarios.
> >
> > [1] and [2] differ fundamentally from HiddenEcho in research goals and application scenarios. DP-FTRL [1] focuses on FL model training optimization, tackling privacy amplification without sampling/shuffling. It suits paradigms where clients deploy full models for local training. Matrix mechanism privacy amplification [2] is a general theoretical framework to tighten privacy bounds, targeting all correlated-noise-based matrix mechanisms but not tailored to LLM-specific traits. HiddenEcho, by contrast, centers on MaaS scenarios, addressing noise amplification in LLM Transformer layers.
>
>
> **If you find our responses helpful in addressing your concerns, we would greatly appreciate your consideration in the scoring!**

---

> > ### Comment · Reviewer_UaRs · 2025-11-21
> >
> > Thanks for the clarification, the responses address my concerns. I have raised the score to 8.

---

> > > ### Author Response · Authors · 2025-11-22
> > >
> > > Thank you very much for your thoughtful feedback. We greatly appreciate your time and constructive engagement with our work!

---

### Author Response · Authors · 2025-12-01
**Summary of Rebuttal**

#### We sincerely thank all reviewers and the AC for their constructive feedback and efforts. Below is a summary of our rebuttal;  full details can be found in the main rebuttal document.

> **Reviewer `UaRs`**
>
> **Concerns**: (1) Communication overhead due to next-token prediction, (2) size of the denoising module, (3) overall communication cost, and (4) comparison with other federated learning methods.
>
> **Response**: We clarify that our engineering implementation leverages separate *prefill* and *decode* phases, combined with server-side caching, to reduce autoregressive transmission burden significantly. The denoising module is lightweight (only tens of MiB), and backward-pass communication costs are minimal. Additionally, we have added comparisons with FL baselines as suggested by the reviewer.
>
> **Result**: All concerns were addressed, and the reviewer raised the score from ``4`` to ``8`` early in the rebuttal period.

> **Reviewer `v3aX`**
>
> **Concerns**: (1) Performance under large-scale models, (2) parameter sensitivity analysis, (3) differential privacy (DP) guarantees for hidden states, and (4) pretraining and selection results of the filter module.
>
> **Response**: We present new experiments on larger models along with comprehensive sensitivity analyses. We further clarify the DP guarantees for hidden states and explain the filter's pretraining procedure, which includes subset generation, distribution shift considerations, and why dynamic re-selection is both challenging and unnecessary.  We also report detailed filter selection results.
>
> **Result**: All concerns were fully addressed.

> **Reviewer `p3hP`**
>
> **Concerns**: (1) Communication cost, (2) terminology clarity, and (3) potential impact of gradient inversion attacks on DP-based privacy guarantees.
>
> **Response**: We provide a detailed clarification of communication costs, revise relevant terminology in the manuscript for precision. We clarified the impact of gradient inversion on DP privacy guarantee, explaining why gradient inversion poses minimal risk to HiddenEcho, which is due to both architectural design and the practical difficulty of such attacks in our setting.
>
> **Result**: All concerns were resolved, and the reviewer increased the score from ``4`` to ``6`` early in the rebuttal period.

> **Reviewer `WawF`**
>
> **Concerns**: (1) Training methodology, (2) communication cost, (3) denoising module size, (4) handling of data distribution shifts in general deep learning settings, (5) inference latency and computational overhead, and (6) experimental interpretation and inclusion of additional variants.
>
> **Response**: We clarify our training strategy and communication efficiency, provide memory usage analysis of the denoising module, discuss robustness to data distribution shifts, and present empirical results on inference latency and computational load. We also expand our experimental analysis and include the variant baseline suggested by the reviewer.
>
> **Result**: All concerns were addressed.

**In summary, we have thoroughly responded to all reviewer comments and will incorporate these clarifications and enhancements into the revised manuscript.**

---

### Meta-Review · Area_Chair_MbcB · 2026-01-07

**Summary:**

This paper proposes a new differentially private split learning algorithm for Large Language Models (LLMs), called HiddenEcho. Specifically, HiddenEcho denoises the client-side embeddings before sending them to the server to improve utility while preserving privacy. HiddenEcho also includes mechanisms to reduce communication overhead.

The reviewers appreciate the following strengths of the paper:

S1. The paper studies an important problem of balancing the trade-off between privacy preservation and model utility in differentially private split learning for Large Language Models (LLMs).

S2. The paper proposes the HiddenEcho algorithm, which is novel and interesting.

S3. The paper provides experiments validating that the proposed method outperforms baseline methods.

The authors have successfully addressed most of the concerns during the rebuttal. In addition, the authors have provided an updated paper that incorporates new experimental results, further discussions, and clarifications. These revisions can be readily incorporated into the camera-ready version.

The ratings were mostly positive before the rebuttal. Reviewer UaRs has explicitly indicated an increase in their rating to 8. Reviewers v3aX and WawF provided positive ratings (6, 6) before the rebuttal and are likely to raise or maintain their positive scores. Reviewer p3hP is likely to raise their rating (from 4 to 6), since the concerns were clarified during the rebuttal.

In summary, the paper has overall positive ratings, with almost all raised concerns addressed, and can be readily incorporated into the camera-ready version. Thus, this paper is a strong candidate for final acceptance.

**Reviewer Concerns:**

The reviewers also raised the following concerns:

W1. Lack of detailed discussion and comparison regarding communication overhead.

W2. Lack of experiments on larger-scale LLMs, parameter sensitivity analysis, additional federated learning baseline methods, and additional variants.

W3. Lack of theoretical analysis of the differential privacy guarantees and lack of clarity in terminology.

W4. Lack of training and inference details.

The authors have successfully addressed most of these concerns during the rebuttal period, as detailed below.

R1. The authors provided additional discussions and empirical results to address W1.

R2. The authors provided additional experimental results to address W2.

R3. The authors provided clarifications and formal differential privacy analysis to address W3.

R4. The authors provided training and inference details, along with further clarifications and new experimental results, to address W4.

**Reviewer Scores:**

The ratings were mostly positive before the rebuttal. Reviewer UaRs has explicitly indicated an increase in their rating to 8. Reviewers v3aX and WawF provided positive ratings (6, 6) before the rebuttal and are likely to raise or maintain their positive scores. Reviewer p3hP is likely to raise their rating (from 4 to 6), since the concerns were clarified during the rebuttal.

---

### Decision · Program_Chairs · 2026-01-26

Accept (Poster)